# RAD-TGTs: high-throughput measurement of cellular mechanotype via rupture and delivery of DNA tension probes

Matthew R. Pawlak [1], Adam T. Smiley [1], Maria Paz Ramirez[1], Marcus D. Kelly[1], Ghaidan A. Shamsan [2], Sarah M. Anderson[2], Branden A. Smeester [3], David A. Largaespada [3], David J. Odde [2] & Wendy R. Gordon [1]✉

Mechanical forces drive critical cellular processes that are reflected in mechanical phenotypes, or mechanotypes, of cells and their microenvironment. We present here "Rupture And Deliver" Tension Gauge Tethers (RAD-TGTs) in which flow cytometry is used to record the mechanical history of thousands of cells exerting forces on their surroundings via their propensity to rupture immobilized DNA duplex tension probes. We demonstrate that RAD-TGTs recapitulate prior DNA tension probe studies while also yielding a gain of fluorescence in the force-generating cell that is detectable by flow cytometry. Furthermore, the rupture propensity is altered following disruption of the cytoskeleton using drugs or CRISPR-knockout of mechanosensing proteins. Importantly, RAD-TGTs can differentiate distinct mechanotypes among mixed populations of cells. We also establish oligo rupture and delivery can be measured via DNA sequencing. RAD-TGTs provide a facile and powerful assay to enable high-throughput mechanotype profiling, which could find various applications, for example, in combination with CRISPR screens and -omics analysis.

Mechanical force has emerged as a critical regulator of cell behavior to drive diverse biological processes from cell migration[1] and stem cell differentiation[2] to discrimination among similar T-cell antigens[3]. Moreover, dysregulation of cellular tensions in disease often leads to different mechanical phenotypes in comparison to normal cells that have been linked to disease progression[4]; breast cancer cells are stiffer[5] while metastatic breast cancer cells are more compliant than normal cells[6]. Mechanotype broadly refers to properties associated with the mechanical microenvironment of a cell and can include quantities such as deformability of the cell and/or the extracellular matrix, cell spread area, cellular contractility and traction force, strength of cell adhesion, and cell migration speed/directionality[7]. Tools to measure cellular mechanotype such as Traction Force Microscopy (TFM)[8] and cell indentation/deformation methods[7,9] lack resolution to probe forces associated with specific ligand-receptor interactions and/or are

inherently low-throughput, requiring high-resolution imaging readouts of tension.

Molecular tension sensors measure the piconewton forces exerted on individual proteins in the cellular context. Numerous variations of molecular tension sensors[10,11] have been developed in the last decade or so, including genetically encoded FRET[11–13] and BRET[14] tension sensors as well as surface-immobilized peptide[15], DNA hairpin[16,17], and DNA duplex-based tension sensors[18,19]. Molecular tension sensors are based on applied force altering the conformation of a "molecular spring" component and an optical readout of the molecular change, such as a change in fluorescence. Molecular tension sensors have been used to study forces sensed by mechanosensors such as integrins[20], cadherins[21], mucins[22], and T-cell receptors[23] and have revealed how tensions are altered when factors in the cellular microenvironment change, such as ECM stiffness[24] or disease-associated mutations[25].

[1]Departments of Biochemistry, Molecular Biology, and Biophysics, University of Minnesota, Minneapolis, MN, USA. [2]Department of Biomedical Engineering, University of Minnesota, Minneapolis, MN, USA. [3]Department of Pediatrics, University of Minnesota, Minneapolis, MN, USA. ✉e-mail: wrgordon@umn.edu

Molecular tension sensors have massive potential to be developed into high-throughput assays that measure changes in cellular mechanical phenotype in combination with CRISPR or drug screens to drive the development of the next generation of mechano-therapeutics.

The Ha lab developed DNA-based molecular tension sensors called Tension-Gauge- Tethers[18] (TGTs), which provide an irreversible and threshold-based readout of tension. TGTs are comprised of DNA duplexes in which one strand is immobilized to a surface and the other strand conjugated to a ligand recognizing a cell surface receptor. When cells are plated on top of TGTs, forces associated with adhesion and migration can mechanically rupture TGT duplexes. A fluorophore or fluorophore quencher pair incorporated into the oligos allows readout of TGT rupture via fluorescence microscopy of the surface, where gain or loss of fluorescent signal is proportional to the total number of accumulated mechanical events over time. Moreover, the tension threshold of DNA duplexes is tunable according to GC content and/or anchoring point of the bottom strand to the surface, allowing measurement of multiplexed forces.

Rapid advances in CRISPR screening and single-cell -omics technology motivate the development of high-throughput mechanotyping technologies, where one could identify genes underlying disease-relevant mechanotypes or perform -omics analysis of cells with a given mechanotype to uncover important genetic pathways. However, molecular tension sensors have several shortcomings which have prevented their use in measuring cellular mechanotype. The key limitation is that rupture of TGTs is typically read out using inherently low-throughput, high-resolution microscopy-based readouts. Recent work from the Salaita group has aimed to adapt TGTs to high-throughput readouts, such as amplification of surface signal that can be measured via plate reader[26,27] and TGTs bound to probe beads allowing for flow cytometry readout of bead fluorescence[28]. However, these methods fail to link the rupture of TGTs to the specific force-generating cell, which prevents their use in mixed cell populations or in selecting/sorting cells of a desired mechanotype.

We present here "Rupture And Deliver" Tension Gauge Tethers (RAD-TGTs) in which flow cytometry is used to record the mechanical history of thousands of cells exerting forces on their surroundings via their propensity to rupture immobilized DNA duplex tension probes. RAD-TGTs can be prepared simply with "off the shelf" oligos by leveraging covalent DNA-to-protein linking HUH-tags[29] to attach desired recombinant protein ligands of interest to unmodified DNA. This conjugation method enabled the use of the broad spectrum and high-affinity integrin ligand echistatin to ensure a robust signal. RAD-TGTs can be plated in commercial 96-well plates with minimal surface preparation. We demonstrate that the flow cytometry readout of fluorescently labeled RAD-TGTs is able to detect the effects of cytoskeletal modulators as well as CRISPR-knockout of relevant cellular mechanosensors. Furthermore, we show that our results can be recapitulated via the sequencing of the internalized oligo, greatly expanding the potential for multiplexed, high-throughput assays such as CRISPR screens. Finally, RAD-TGTs can be used to analyze mixed populations of cells and distinguish them by mechanotype.

## Results

### RAD-TGTs: design of "off-the-shelf" rupture and deliver DNA tension probes leveraging HUH-endonucleases

The goal of this study was to develop a high-throughput assay to profile the mechanotype of cells based on how they physically interact with their environment. We reasoned that existing immobilized TGT DNA tension probes[18] could be converted into a high-throughput platform by changing the focus of the readout from the oligo remaining on the surface to the oligo captured and internalized by the cell upon rupture (Fig. 1a). Indeed, early studies of TGTs noted that fluorescent oligos appeared to be internalized by the force-generating cell[30]. We hypothesized that tension-dependent rupture of TGTs and

subsequent delivery of the ruptured oligo into the cell of interest could be detected by flow cytometry or next-generation Sequencing. This would allow for rapid recording and quantification of the cumulative force-dependent rupture events that cellular forces generate over a period of time in the cell of interest. We expect that just as cells exert different traction forces on their surroundings depending on their mechanical properties, cells will have different propensities to rupture TGTs, thus resulting in a high-throughput measurement of cellular mechanotype.

RAD-TGTs were adapted from the TGT designs originally reported by the Ha group[18], which consist of two strands of DNA—one that is conjugated to a ligand (the ligand strand) and a strand that contains a modification to immobilize the duplex (the anchor strand) (Fig. 1a). The duplex component of the TGT remains identical to previous designs so as not to alter the thermodynamic properties of DNA duplex formation. The anchor strand contains a biotin modification either at the 5′ or 3′ end that binds plated neutravidin to yield TGTs that rupture at different forces either in the unzipping (12pN) and shear (54pN) modes, respectively. For readout of rupture and delivery into cells, the ligand strand also contains a 3′ fluorophore or a short nucleotide barcode depending on the choice of readout. Furthermore, the anchor strand contains a 5′ quencher to control for spurious detachment of the entire duplex from the surface, which may occur during cell handling (Supplementary Fig. 1). Detached duplexes would not be fluorescent in the presence of the quencher such that signal only reports on the rupture of the DNA duplex.

Typically, the ligand strands of TGTs are chemically modified with a ligand such as integrin-binding cyclic-RGD using amino, thiol, or Click chemistry. Our design leverages DNA-to-protein linking HUH-endonucleases to allow covalent attachment of versatile protein-based ligands to RAD-TGTs. Briefly, viral HUH-endonucleases are small (<20 kDa) proteins that can be fused to ligands of interest to form robust, sequence-specific phospho-tyrosine covalent bonds with a short nona-nucleotide sequence of ssDNA within minutes under physiologic conditions[29]. Thus the ligand strand does not require chemical modification as in other TGT designs but simply includes a short 5′ DNA extension. The majority of TGT studies have used integrin-binding linear or cyclic-RGD peptides due to the relative ease of chemical conjugation to DNA. These peptides have a broad range of affinities (low nM to µM)[31] for different integrin subtypes and result in an estimated 1000–10,000 rupture events per cell[26], which typically requires high NA objectives and high sensitivity cameras to image. We aimed to broaden the spectrum of ligands accessible to TGT studies by conjugating protein ligands of interest to TGTs via ligand-HUH fusion proteins. To benchmark our studies against prior TGT work, we expressed a fibronectin domain (FN)[32] in fusion with the HUH-tag derived from the Wheat Dwarf Virus (WDV)[33] with expected integrin-binding activity similar to RGD domains. To enhance cellular interaction with TGTs and increase rupture events, we also expressed echistatin (Echi) fused to WDV. Echi is a small toxin-derived broad-spectrum integrin-binding protein presenting the RGD peptide in a tight loop with low- to sub-nanomolar affinity for every subtype of RGD integrin tested[31]. The resulting proteins (now known as WDV-Echi or WDV-FN) can be covalently tethered to annealed duplex TGTs prior to immobilization (Supplementary Fig. 2). To account for any background signal from non-ligand-mediated rupture events, WDV conjugated TGTs are used as a negative control in all experiments.

### CHO-K1 cells plated on RAD-TGTs exhibit expected cell adhesion and surface fluorescence characteristics

We first aimed to assess whether RAD-TGTs functioned similarly to TGTs in previous studies. The TGT function of RAD-TGTs was confirmed by recapitulating the original adhesion assays performed by Wang and Ha[18]. These experiments demonstrated that CHO-K1 cells

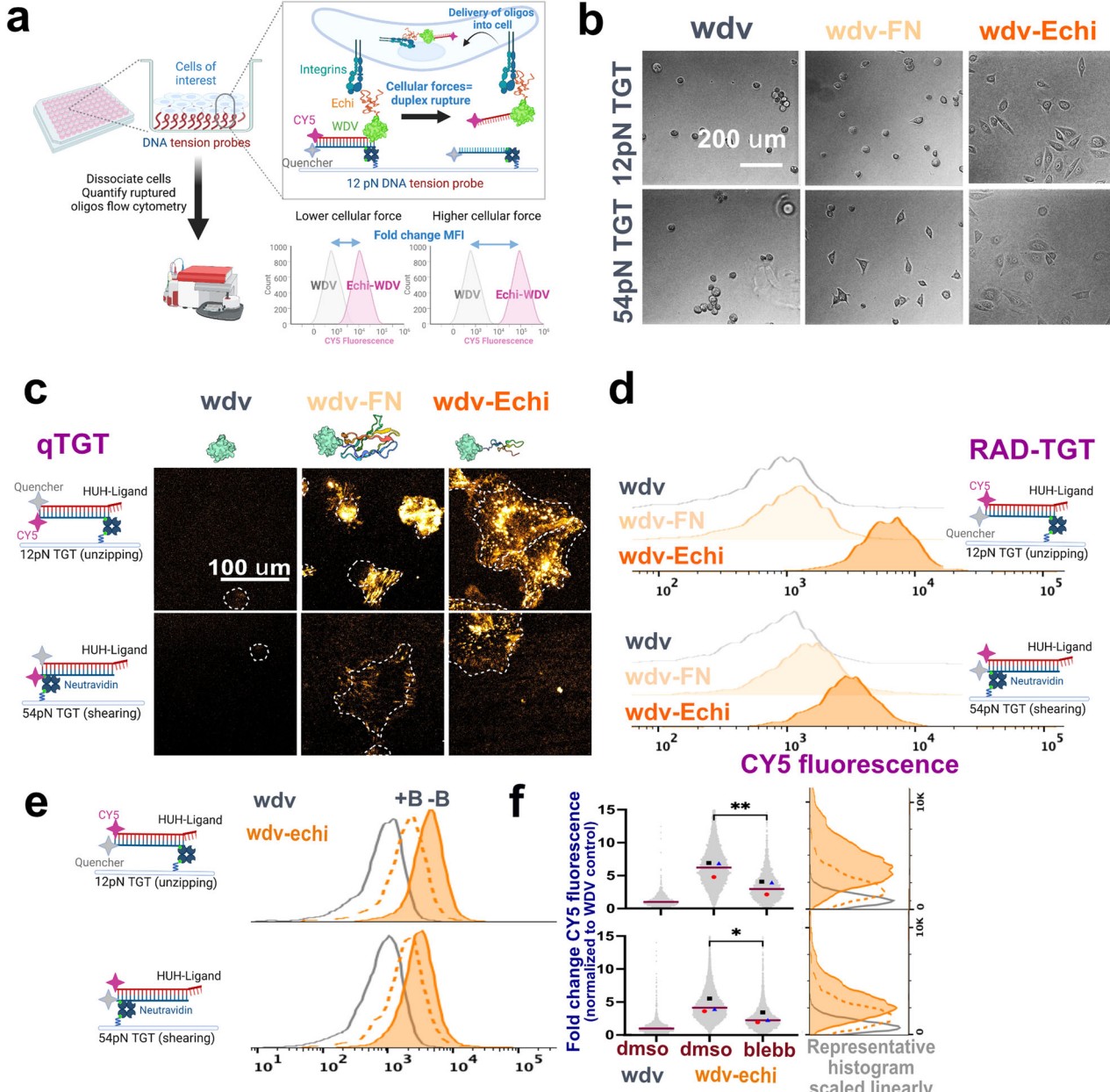

**Fig. 1 | Overview and validation of RAD-TGT function in CHO-K1 cells.**
**a** Conceptual schematic of the force-induced rupture and readout of RAD-TGTs.
**b** Brightfield imaging at 20x of adhesion assay of cells on RAD-TGT surfaces with either WDV, WDV-FN, or WDV-Echi ligand and 12 or 54 pN rupture force.
**c** Fluorescent imaging at 40x of qTGT fluorescent duplex rupture of cells plated on TGT surfaces of varying ligand and rupture force composition. White dotted lines denote cell borders. **d** Flow cytometry results of cells plated on RAD-TGT surfaces with different ligand and rupture force composition. **e** Representative histogram showing the effect of para-amino-blebbistatin treatment on TGT rupture.

**f** SuperPlots of CY5 fluorescence of each cell from three biological replicates normalized to WDV median fluorescence with symbols for medians of biological replicates. Representative histograms turned 90° and scaled linearly for reference. Gray histograms are WDV only, dotted lines are with para-amino-Blebbistatin, and solid orange is with DMSO treatment as control. **p = 0.001387, *p = 0.015441. Statistics were performed using one-way ANOVA of the medians of biological replicates. Source data are provided as a Source Data file. All cartoons and schematics were created with Biorender.com.

failed to adhere to surfaces coated with 12 pN TGTs but could adhere to surfaces coated with 54 pN TGTs, suggesting that adhesion requires integrins to exert more than 12 pN of force. These results were recapitulated using the WDV-FN ligands conjugated to RAD-TGTs (Fig. 1b). Interestingly, CHO-K1 cells adhered to both 12 and 54 pN WDV-Echi surfaces. This is likely attributed to the higher affinity of echistatin for integrins and/or the wider array of integrin subtypes bound by echistatin than cyclic-RGD or FN, which may lead to adhesion in different force regimes as was demonstrated for α4β1 integrins in recent studies[34]. Nevertheless, these results indicate that the assembled TGTs

function as expected and that surface preparations are equivalent to prior studies.

We next measured TGT rupture via a traditional fluorescence microscopy readout using quenched TGTs (qTGTs). Briefly, qTGTs are similar to RAD-TGTs, but the fluorophore and quencher locations are inverted, thus allowing for a gain of fluorescence on the surface of the plate following rupture. As expected, CHO-K1 cells plated on 12 and 54 pN qTGTs conjugated to WDV adhered poorly and resulted in no fluorescence signal (Fig. 1c, Supplementary Fig. 3). Cells plated on the WDV-FN qTGTs resulted in rupture patterns similar to previous

studies. Briefly, cells on the 12 pN surface struggled to adhere but caused robust rupture uniformly across the small cell footprint, while cells on 54 pN qTGTs adhered well but ruptured qTGTs in a streak pattern caused by motile focal adhesions[30] (Fig. 1c). This behavior corresponds to a higher rupture ratio of 12pN TGTs but an overall lower cumulative fluorescence intensity given the small cell footprint compared to 54pN TGTs, as corroborated by quantification of cumulative fluorescence intensity (Supplementary Fig. 3). In previous studies, whether the integrated fluorescence is higher for 12pN versus 54pN TGTs depended on how well cells adhere to 12pN TGTs (Supplementary Table 1). Intriguingly, cells plated on 12 and 54 pN TGTs conjugated to WDV-Echi, where cells adhere similarly, show both streaks and fluorescent puncta with a higher cumulative fluorescence signal across the cell footprint for 12pN than 54pN TGTs (Supplementary Fig. 3).

### Flow cytometry detects rupture and delivery of immobilized RAD-TGTs

We next asked if internalized fluorescence derived from TGT rupture could be detected by flow cytometry (Fig. 1d). Briefly, assembly of the RAD-TGTs (see "Methods" for full details) involves: (1) coating standard glass-bottom 96-well plates with biotin-BSA, (2) incubation with neutravidin, and (3) incubation with 80 µl of a 1 µM solution of assembled TGTs overnight. After washing, approximately 15,000 cells per well are plated in serum-free media to mitigate DNA degradation by nucleases for 1–4 h. At the desired time point, the cells are trypsinized and analyzed by flow cytometry.

CHO-K1 cells were plated on both 12 and 54 pN RAD-TGT surfaces and then analyzed after 90 min incubation (Fig. 1d). Two measures are typically used to quantify TGT rupture using imaging; rupture ratio and integrated intensity for a given cell footprint; we would expect flow cytometry to mirror integrated fluorescence measurements. Flow cytometry of the CHO-K1 cells shows modest increases in median fluorescence for the WDV-FN ligands compared to WDV alone. The 12pN TGT showed lower fluorescence than 54 pN, likely due to the lack of cell adhesion under these conditions, as has been observed previously[35]. Generally, literature studies show 12 pN TGTs are ruptured more readily than 54 pN TGTs fluorescence, but only under conditions where cell adhesion is facilitated by mixing 12pN TGTs with TGTs of higher tension tolerance[36] or by coating surfaces with fibronectin[37]. Indeed, when WDV-Echi was conjugated to TGTs where cells adhere without assistance on both 12 and 54 pN TGTs, the fold change of CY5 median fluorescence intensity compared to WDV control was threefold and sevenfold for the 54 and 12 pN RAD-TGTs, respectively. The fluorescence histograms observed from rupture and delivery of RAD-TGTs are narrow, with distinct shifts in median fluorescence of the population compared to WDV alone negative controls. In contrast, other studies detecting cellular forces by flow cytometry of microparticles containing DNA tension probes interacting with cells of interest instead report a modest broadening of the population in response to cellular tension[28]. This suggests that a homogeneous population of cells plated on the RAD-TGTs exerts tension on the surface.

To determine whether the oligo was fully internalized or if some remained bound to the surface of the cell, we utilized a common assay in assessing DNA origami delivery[38]. Briefly, cells were plated on RAD-TGTs. Following incubation, cells were treated with a promiscuous nuclease or vehicle control and then analyzed via flow cytometry. There was a slight decrease in fluorescence following nuclease treatment (Supplementary Fig. 4), indicating that indeed a large percentage of the oligo delivered to the cell (~90%) was indeed internalized. Additional characterization and optimization of the RAD-TGT protocol confirmed that internalized fluorescence increases with time after cell plating up to about 2 h (Supplementary Fig. 5) and that overall median fluorescence does not depend strongly on cell density (Supplementary

Fig. 6). We also performed measurements including fibronectin in the coating protocol to ensure the results are not confounded by differences in cell behavior following TGT rupture (Supplementary Fig. 7), as other studies have done (Supplementary Table 1). Generally, all surface preps result in the same trends in the data and cells adhering well after 2 h with varying morphologies as expected. It should be noted that experiments were attempted on commercially available neutravidin-coated polystyrene plates, and the resulting fluorescence profiles showed non-specific adsorption and volatile and noisy readouts (Supplementary Fig. 8). Thus, all experiments performed in this study are on 96-well glass-bottom plates.

### RAD-TGT signal is significantly decreased by blebbistatin treatment

Finally, previous TGT studies in a variety of cell lines (Supplementary Table 1) have demonstrated significant decreases in TGT rupture in the presence of cytoskeletal inhibitors, such as the MyosinII inhibitor para-amino-blebbistatin[39]. Indeed, blebbistatin treatment decreased median fluorescence for both 12 and 54pN TGTs, as shown in the representative fluorescence histograms (Fig. 1e). In order to illustrate the reproducibility of RAD-TGT experiments, we also plot the fluorescence of every cell from three biological replicates (5–10k cells) in gray (Fig. 1f). Each WDV-Echi data set is then normalized to its corresponding WDV alone dataset, to yield a fold change of CY5 fluorescence. The medians of the three biological replicates are shown as symbols on top of the individual data points in SuperPlot style[40]. We also show the representative histograms of the corresponding SuperPlot data plotted on a linear scale for comparison. Overall, blebbistatin decreases the median fluorescence of both 12 and 54 pN by about 50%, which is in line with literature observations (Supplementary Table 1). The decrease in signal following drug treatment indicates that the TGT rupture and subsequent internalization into the cell depend on cellular forces. Intriguingly cell spread area slightly increased following treatment even as RAD-TGT signal decreased, which has been observed previously[41] but indicates that changes in fluorescence as a result of rupturing RAD-TGTs is not simply a measure of cell spread area (Supplementary Fig. 9).

Together, all of the experiments thus far establish the high-throughput flow cytometry readout of thousands of cells in minutes faithfully recapitulates prior experiments using TGTs. Moreover, these studies demonstrate that the use of Echistatin as a ligand drastically increases signal and allows cells to spread well on both 12pN and 54pN TGTs without the need for additional cell adhesion agents.

### Cellular rupture of RAD-TGTs can also be detected by sequencing DNA barcodes

We next expanded our studies to U251 glioma cells which are derived from an extremely malignant and invasive tumor, to determine their propensity to rupture RAD-TGTs. As with CHO-K1 cells, U251 cells did not adhere well to TGTs conjugated with WDV-FN (Supplementary Fig. 10). The fold change in median fluorescence was under twofold and not statistically significant, though our use of the median fluorescence to determine statistical significance is more stringent than other published methods which use %positive cells above an arbitrary gate (Supplementary Fig. 11). On WDV-Echi ligands, U251 cells generated signals much greater than CHO-K1 cells, with the 12 and 54 pN RAD-TGTs resulting in 12-fold and sevenfold increases in median fluorescence, respectively (Fig. 2a, b). Due to the robust signal, we used WDV-Echi as our ligand of choice unless otherwise noted.

To confirm the differences observed between 12 and 54 pN TGTs intensity, we performed qTGT imaging, which also showed higher fluorescence intensity with 12 pN WDV-Echi qTGTs than the 54 pN qTGTs (Fig. 2c). As expected, rupture of both 12 and 54pN TGTs was decreased upon treatment para-amino-Blebbistatin (Supplementary Fig. 12), though the decrease for 54pN was not statistically significant.

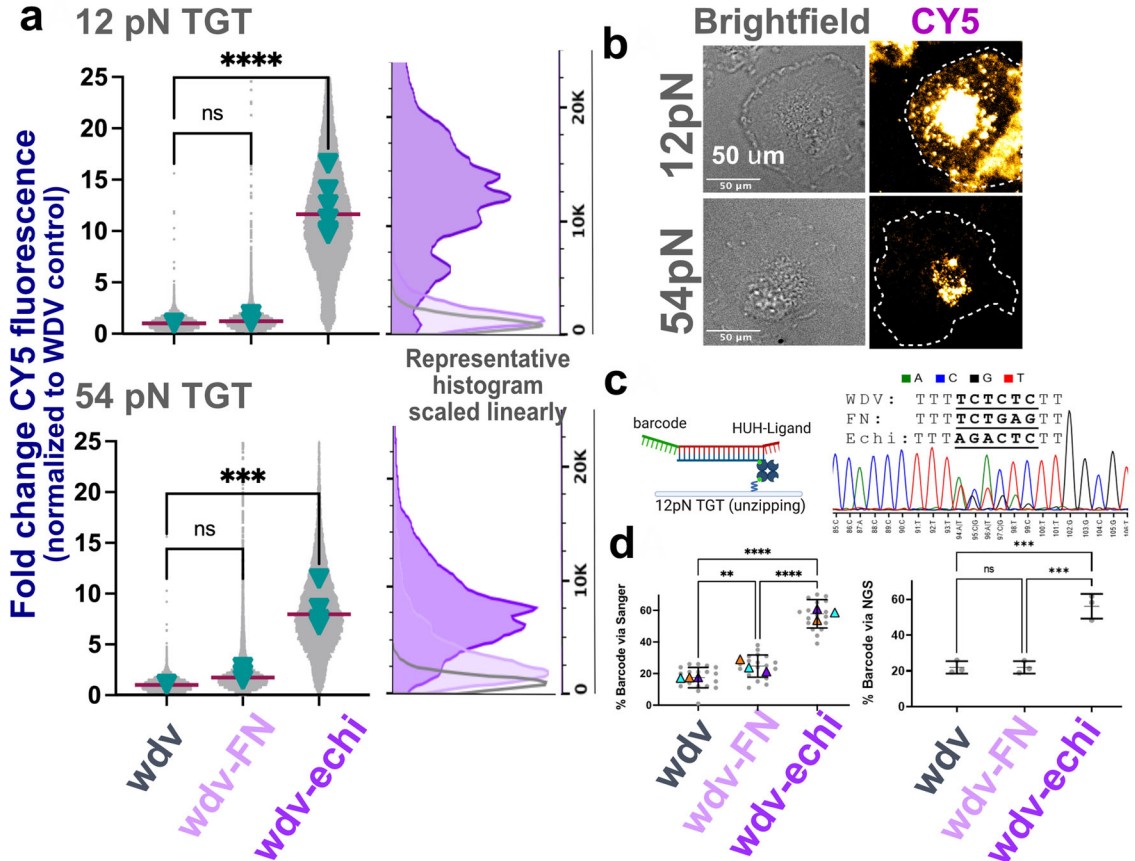

**Fig. 2 | RAD-TGT measurements in U251 cells readout by flow cytometry and sequencing of internalized DNA barcodes. a** SuperPlots of the fold change of CY5 fluorescence intensity of individual U251 cells plated on 12 pN (top) or 54 pN (bottom) RAD-TGTs with either WDV, WDV-FN, or WDV-Echi. Triangles represent the median fold change of population per replicate; the horizontal red line is at the median fold change for all cells analyzed. *n* = 8 or 5 independent experiments for 12 pN and 54 pN graphs, respectively. ***p = 0.0002, ****p < 0.0001. **b** Brightfield and qTGT imaging of U251 cells plated on either 12 pN or 54 pN qTGTs with WDV-Echi ligand. **c** Schematic of 12 pN RAD-TGT with barcode used for sequencing experiments and representative Sanger sequencing trace with each ligand barcode. **d** Sanger and next-generation sequencing results of biological triplicates of U251 cells plated on an equimolar mixture of 12 pN RAD-TGTs with all three ligands present. **p = 0.0077; ****p < 0.0001. Source data are provided as a Source Data file. All statistics were performed using a one-way ANOVA of the medians of biological replicates. Error bars are standard deviations.

This could be in part due to the rigor of using median fluorescence intensity to compare populations or that microtubule dynamics play a substantial role in U251 traction force generation[42]. To ensure this increase in signal was not due to any differences in membrane-bound nuclease activity between cell lines, as has been observed for some cancer cell lines, we used a surface-bound nuclease sensor to visualize nuclease activity[43]. We observed that U251 cells exhibit very little nuclease activity compared to force-ruptured TGTs (Supplementary Fig. 13). Furthermore, U251 cells had similar nuclease activity to CHO-K1 cells which have previously been characterized as having low nuclease activity[43].

The traction forces generated by cells have been shown to correspond to the numbers of "clutches" linking the actin cytoskeleton to the extracellular matrix, e.g., integrin-anchored complexes[44,45]. To assess differences in RGD-binding integrins on the different cell types, we titrated CHO-K1 and U251 cells with soluble fluorescent oligos conjugated to Echi. We observed U251 cells bound three- to fourfold more than CHO-K1 cells, suggesting a greater number of integrin clutches available to exert traction force (Supplementary Fig. 14).

We envisioned that the utility of RAD-TGTs could be expanded beyond flow cytometry using DNA sequencing methods such as next-generation sequencing. As proof of concept, we appended unique DNA barcodes to the 3′ end of the ligand strand (Fig. 2d) and conjugated the resulting 12pN RAD-TGTs to WDV, WDV-FN, and WDV-Echi. We immobilized them in the same well (resulting in 1/3 the normal

concentration of each flavor of TGT) and performed the experiment with U251 cells as usual. We then amplified the internalized TGTs from cell lysates and sent the samples for both Sanger and Illumina next-generation sequencing. We also set up a reference experiment with the same composition of ligands or one ligand per well but with fluorescent oligos so that we could obtain analogous flow cytometry results (Supplementary Fig. 15). We used web-based sequence deconvolution programs to calculate the percent of each barcode represented in the samples quantified by Sanger sequencing and Biopython to quantify NGS data (Fig. 2d). Remarkably, the quantification of ruptured TGTs by sequencing corresponds with the trends seen in the analogous flow cytometry experiments (Supplementary Fig. 15), in which WDV-Echi provides a very robust signal relative to the co-plated ligands WDV and WDV-FN. This recapitulation of results via sequencing indicates that, indeed RAD-TGTs can be analyzed via sequencing.

## Mechanotype is altered when putative mechanosensing proteins are knocked out

The high-throughput mechanotype readout resulting from RAD-TGT rupture could enable CRISPR screening for the knockout of cellular proteins that alter cellular force generation. As proof of concept, we compared the propensity of wildtype U251 cells (WT) to rupture RAD-TGTs against U251 cells with CRISPR knockouts of two putative mechanosensing proteins—talin-1 and CD44 (Fig. 3a). Talin-1 is a mechanosensor in the integrin-anchored focal adhesion complex. Loss

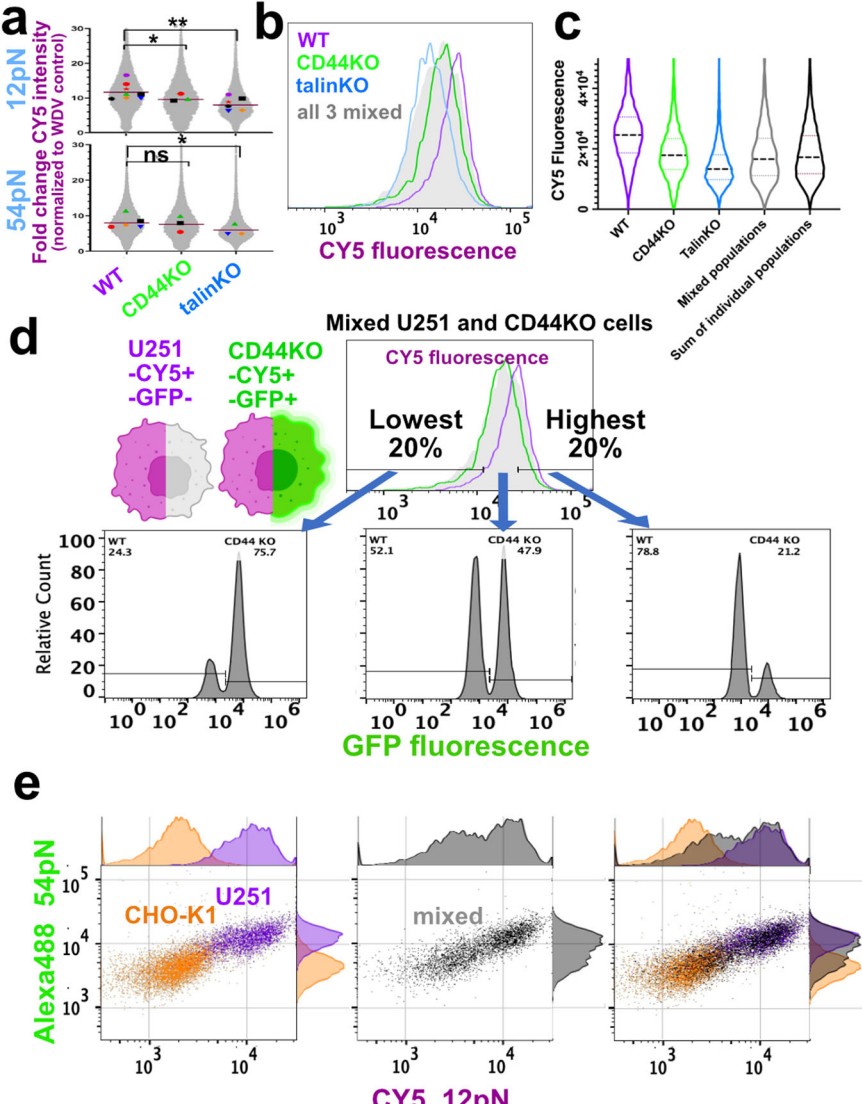

**Fig. 3 | RAD-TGTs detect CRISPR-KO of mechanosensing proteins in single and mixed cell populations. a** SuperPlots of the fold change of CY5 fluorescence intensity of individual WT, TLN1 KO, or CD44 KO U251 cells on 12 pN (top) or 54 pN (bottom) RAD-TGTs with WDV-Echi. Symbols represent the median Cy5 fold change for each replicate experiment. Identical symbols indicate the samples were collected on the same day. The horizontal red line is at the median fold change for all cells analyzed. For 12 pN experiments, $n = 8$ (WT), 3 (CD44KO), or 6 (talinKO) and for 54 pN experiments $n = 5$(WT) and 3 (CD44KO and talinKO). A two-tailed paired $t$-test was used for statistics, *$p = 0.0360$ (12 pN CD44KO), **$p = 0.0028$ (12 pN talinKO), and *$p = 0.0322$ (54 pN talinKO). Medians were paired based on the day each experiment was performed. **b** Histogram of a mixed population of U251 WT,

Talin1, and CD44 KO (gray solid), overlaid with histograms of cell lines tested individually (outlines only). **c** Violin plots of individual populations, mixed population, and the sum of individual populations. **d** Cartoon of U251 WT and CD44 KO cells and accompanying histogram of the two cell types mixed together (gray) or individual. The composition of the population is determined by measuring the GFP signal on different applied gates of the CY5 histogram as CD44 KO also expresses GFP. The composition of the lowest 20%, highest 20%, and the entire population is displayed. **e** Scatter plots of U251 and CHO-K1 cells plated on surfaces containing a 12 pN Cy5 labeled RAD-TGT and 54 pN A488 labeled RAD-TGT for both each cell individually and in a mixed population. Source data are provided as a Source Data file. All cartoons and schematics were created with Biorender.com.

of talin-1 alters cell spreading and focal adhesion formation but does fully ablate activity due to the compensatory functions of talin-2[46]. CD44 displays mechanosensitive behavior in glioma cells[47] and is also thought to participate in crosstalk with integrin-based focal adhesions via common binding to the actin cytoskeleton[48,49]. U251 talinKO cells displayed a rounded morphology on TGTs, while CD44 cells resembled wildtype U251 cells (Supplementary Fig. 16). Knockout of both mechanosensing proteins reduced the median fluorescence upon rupture of 12pN TGTs by 25–30% (Fig. 3a), with talin having a slightly larger effect on TGT rupture. Interestingly, differences in mechanical phenotype emerged when knockout cell lines were plated on 54pN TGTs. Talin-KO also significantly decreased cellular force exerted on 54pN TGTs (~35% decrease in fold change). However, the effect of

knockout of CD44 was blunted on 54 pN RAD-TGTs (~15% decrease), revealing potentially interesting mechanistic differences in the roles that talin and CD44 play in force sensing and also revealing that multiplexing of TGTs with different tension tolerances can provide new mechanistic insights, as others have demonstrated.

## RAD-TGTs can detect multiple cell populations in 1D and 2D histograms

We noted that the RAD-TGT fluorescence histograms of the three cell lines resulted in distinct, separated peaks (Fig. 3b). We plated a mixture of the three cell lines and observed that the fluorescence envelope resembles the sum of the three individual populations (Fig. 3b). Indeed, violin plots of the cumulative fluorescence plots of the three

individual populations closely resemble the mixed cell population (Fig. 3c) We next wondered if gating on cells at the high or low end of a mixed cell fluorescence profile would enrich for expected cells based on their mechanotype. We took advantage of the fact that the CD44KO cells express a GFP transgene. We mixed WT and CD44KO cells and plated them on 12pN RAD-TGTs (Fig. 3d). As expected, the resulting fluorescence profile of the mixed population (Fig. 3d) recapitulates the two underlying cell populations. We then gated on the highest and lowest 20% of cells on the edges of the mixed cell fluorescence profile and looked at the GFP± populations. As expected, there are equivalent high and low GFP populations for the entire CY5 profile. The 20% of cells exhibiting the lowest CY5 fluorescence is comprised of a population enriched for high GFP, which corresponds to CD44KO cells. In contrast, the top 20% of fluorescent cells are enriched in the low GFP WT U251 cells. These inquiries demonstrate the potential of using RAD-TGTs in pooled CRISPR screens to sort cells with mechanotypes distinct from parent cells to identify genes that may contribute to that mechanotype. These results corroborate our initial findings of CD44 KO exerting decreased forces on DNA tension probes compared to wildtype while also demonstrating that we can identify cells with altered mechanotype in a mixed population.

We next measured profiles of mixed populations of cells on a mixture of 12 and 54 pN TGTs, resulting in 2D dot plots (Fig. 3e) to potentially improve population separation and resolution. To do so, we prepared surfaces containing a 12 pN Cy5 labeled RAD-TGT and a 54 pN Alexa488 RAD-TGT with the rationale that having two dimensions should increase the resolution and dimensionality of the mechanotype. We tested this system with WT U251 cells and CHO-K1 cells either individually or in the context of a mixed population. Much like the 1D plots, the mixed population of cells appeared as a composite of the two cell lines (Fig. 3e). Furthermore, the experiment was repeated with fluorophores swapped between RAD-TGTs, and results were maintained (Supplementary Fig. 17). The 2D plot serves as a promising next step in RAD-TGT development and usage.

## Discussion

We present a high-throughput platform called RAD-TGTs that uses flow cytometry to record the mechanical history of a cell *in the cell of interest* via the rupture of immobilized DNA duplex tension probes. The force exerted by cells on their surroundings is one commonly measured parameter of mechanical phenotype or mechanotype[8]. Thus, this assay has the potential to screen for drugs or genes that alter mechanotype in a high-throughput manner. After validating that RAD-TGTs recapitulate prior work, we demonstrate that rupture of RAD-TGTs depends on cell type; U251 cells promote more rupture of TGTs than CHO-K1 cells, likely due to increased integrins measured on the cell surface. We also establish that the rupture of RAD-TGTs can be read out in the cells of interest by sequencing the internalized DNA strand, with DNA barcodes substituting for the fluorophore. As proof of concept of using RAD-TGTs in CRISPR-KO screens, we show that CRISPR-knockout of putative mechanosensors talin-1 and CD44 results in unique mechanotypes, distinguishable from wildtype and each other. Finally, we show that fluorescence profiles of mixed populations of cells manifest as the sum of the individual cell populations, underscoring the potential of the assay to screen on mechanotype; cells with high or low fluorescence on the profile edges are enriched for the expected cell population based on mechanotype.

The assay is based on well-studied DNA tension probes, originally known as Tension Gauge Tethers (TGTs)[18], in which rupture of the duplexes by cells is read out by high-resolution imaging on the surface. Signal strengths have been further improved in the form of qTGTs[50] and Integrated Tension Sensors (ITSs)[51], for example. Previous attempts have been made to adapt TGTs to high-throughput readouts, such as amplifying the signal of the surface-bound anchor strand after duplex rupture such that it can be read out on a plate reader[26,27], as well

as immobilizing TGTs on microparticle probe beads[28] and performing flow cytometry analysis on them for interrogation of cellular forces. RAD-TGTs have four distinct features that distinguish them from other DNA tension probe systems: (1) The main advance of our assay is the readout of TGT rupture in the cell of interest; this allows the mechanical history to remain with the cell that exerted forces, enabling combination with techniques such as pooled CRISPR screens and -omics analysis of cells with distinct mechanotypes, (2) the use of DNA-to-protein linking HUH-endonucleases to easily attach protein-based ligands to TGTs without chemical steps or DNA purifications, (3) the use of a broad spectrum and high-affinity integrin ligand echistatin to boost the signal, and (4) high-throughput, highly multiplexable readouts of mechanical history via next-generation sequencing (in addition to flow cytometry).

Here we demonstrate that RAD-TGTs report significant differences in signal following changes in cellular-generated forces. However, other factors outside of direct force application through integrins may alter signals, such as cell spread area, variability in receptor expression/ degradation, ligand affinity, endocytosis efficiency, and ECM remodeling between cells. These factors likely all contribute to observed cellular mechanotypes. Moreover, the fluorescence could also be influenced by factors unrelated to cellular forces, such as cell size. Thus, caution should be taken in interpreting differences in fluorescence signals between different cell types. Nevertheless, RAD-TGTs offer a rapid screen to identify putative mechanotype modulators, which can be further validated with mechanistic studies.

RAD-TGTs offer distinct advantages over established methods to probe mechanotype, such as traction force microscopy (TFM), which involves measuring to what extent cells deform polymer gels embedded with fluorescent beads[8,52]. However, TFM is limited by its resolution of nN forces, the requirement for high-resolution imaging, and its complex data analysis, limiting its use as a high-throughput mechanotyping assay. The principle of RAD-TGTs in recording forces exerted by cells on a surface is similar to TFM, though with several notable differences: first, TGT rupture is irreversible, and a cell accumulates ruptured TGTs the entire time of the experiment. TFM provides more of a snapshot at a given time since the deformability is reversible; second, TFM is only sensitive to contractile forces that can deform the polymer. TGTs are much more sensitive and measure the sum of pN scale molecular events. TGTs are also tunable and can reveal forces of different scales exerted by cells.

For instance, we showed that talin and CD44 knockouts in U251 cells decreased the rupture of 12pN TGTs to similar extents. While talinKO also significantly reduced 54pN rupture, CD44KO did not. Talin's role in facilitating traction force in cells is relatively well understood, given its direct role in focal adhesion dynamics, but CD44 is less studied. CD44 is an extracellular matrix receptor that connects to the actomyosin machinery via adapter proteins like integrins, and crosstalk with integrins has been reported[53,54]. The data suggest that CD44 may be more involved in the formation of nascent focal adhesions and initial cell adhesion/spreading and less in mature focal adhesions and cell contractility, supported by prior studies demonstrating that CD44KO does not disrupt focal adhesion formation[55]. Moreover, CD44 is thought to act as a "picket" in the cell membrane "fence", which coordinates receptor organization[56]. Thus the loss of CD44 leads to the disorganization of cell surface receptors, which may alter initial adhesion events and subsequent 12 pN rupture events.

The use of HUH-endonucleases in RAD-TGTs expands the repertoire of proteins that can be attached to TGTs to allow probing of a broader range of receptor–ligand interactions, adaptation to more complex microenvironments such as 3D hydrogels, and expansion of readouts possible after rupture and delivery. For example, the "anchor" strand of TGTs could be conjugated via HUH-tags to collagen binding CNA35[57] to allow immobilization of TGTs in collagen gels.

The ligand strand could be conjugated to luciferases, nanobodies, or even genome engineering reagents to allow readout of TGT rupture in animals or tension-dependent delivery of reagents.

RAD-TGTs in their current form should allow the study of cellular processes in many cell types but can be improved in future studies. The readout was sensitive enough to detect oligos conjugated to a single fluorophore in multiple cell lines without requiring additional amplification steps used in recent studies[26]. However, the sensitivity varies among cell lines, depending on adhesive and traction forces generated; thus, the flow cytometry readout may not be suitable for cell lines that generate low traction force. The sensitivity of the flow-based readout could be improved by attaching tandem fluorophores or even quantum dots[58]. Another concern of all DNA-based sensors, including RAD-TGTs, is the effect of nucleases on sensor stability and readout. Although the cell lines we tested had minimal nuclease activity, it has been known that some cancer cell lines have elevated nuclease activity[43]. Fortunately, recent work has demonstrated that PNA-DNA hybrids are not cleaved by surface nucleases[59]. Future iterations and improvements of RAD-TGTs may include the use of a PNA bottom strand to allow for use with cells that have high nuclease activity.

Finally, the RAD-TGTs presented in this study were all assembled using oligos purchased from IDT, with no additional chemical modifications or purifications in our lab, and ligands containing HUH-fusions can be easily prepared using *E. coli* expression protocols. Moreover, commercial 96-well plates were used with minimal surface preparation. Thus, the advances offered by RAD-TGTs should make DNA tension sensor probes accessible to more research labs.

## Methods
### HUH-ligand preparation
HUH variants were expressed in *E. coli* and purified following previously described protocols[29,33]. Briefly, proteins were expressed in BL21(DE3) *E. coli* following IPTG induction. *Escherichia coli* were lysed, and protein was purified via subsequent Ni-NTA affinity chromatography and size exclusion chromatography.

### RAD-TGT synthesis
TGTs were designed using previously characterized DNA sequences[51] with a 5′ extension to allow for HUH binding. The following oligonucleotides were purchased from Integrated DNA Technologies.

### Fluorescently labeled ligand strand
5-GCT ATA AAC TCA CCG TAA TTT TTT GGC CCG CAG CGA CCA CCC TTT/3Cy5Sp/-3

### Quencher ligand strand
5-GCT ATA AAC TCA CCG TAA TTT TTT GGC CCG CAG CGA CCA CCC TTT/3IAbRQSp/-3

12 pN Quencher Anchor Strand: 5-/5IAbRQ/GGG TGG TCG CTG CGG GCC/3Bio/-3

12 pN Unlabeled Anchor Strand: 5-GGG TGG TCG CTG CGG GCC /3Bio/-3

12 pN Fluorescently Labeled Anchor Strand: 5-/5Cy5/GGG TGG TCG CTG CGG GCC /3Bio/-3

54 pN Quencher Anchor Strand: 5-/5IAbRQ/iBiodT/GGG TGG TCG CTG CGG GCC-3

54 pN Unlabeled Anchor Strand: 5-/5Bio/GGG TGG TCG CTG CGG GCC-3

54 pN Fluorescently Labeled Anchor Strand: 5-/5Cy5/iBiodT/GGG TGG TCG CTG CGG GCC-3

Barcoded Ligand Strand 1: 5-GCT ATA AAC TCA CCG TAA TTT TTT GGC CCG CAG CGA CCA CCC TTT TCT CTC TT GGC GTC ATC GTG TAC CGG-3

Barcoded Ligand Strand 2: 5-GCT ATA AAC TCA CCG TAA TTT TTT GGC CCG CAG CGA CCA CCC TTT AGA CTC TT GGC GTC ATC GTG TAC CGG-3

Barcoded Ligand Strand 3: 5- GCT ATA AAC TCA CCG TAA TTT TTT GGC CCG CAG CGA CCA CCC TTT TCT GAG TT GGC GTC ATC GTG TAC CGG-3

Forward Amplicon Primer: 5-ACA CTC TTT CCC TAC ACG ACG CTC TTC CGA TCT CTC ACC ATG TGG TGA CGC CAG AAT TTG CCG CAA TAC ACA GTT TAC GCC GTT CGG TCA GCT TGG TAT CCG TAG CGC AGC GAC CAC CCT TT-3

Reverse Amplicon Primer: 5--GAC TGG AGT TCA GAC GTG TGC TCT TCC GAT CTG AGA AAC CTA AGC AGA CTT CTC CTG GTC GAT GAT TGA TAA GGG TCT CGG AAT GTC CCC GGT CGC ATG GTT CCG GTA CAC GAT GAC GCC-3

To prepare TGTs, anchor and ligand strands were mixed in a 1.1:1 ratio and then annealed in a 1x annealing buffer (10 mM Tris pH 7.5, 50 mM NaCl, 1 mM EDTA) by heating at 98 °C for 5 min followed by cooling at room temperature for 1 h. The excess bottom strand was used to mitigate any single-stranded fluorescent top strand which may be internalized, resulting in false positives. RAD-TGTs were then generated by reacting the HUH-ligand of interest with the annealed duplex in a 2:1 ratio. Reactions were performed in the following buffer (50 mM HEPES pH 8.0, 50 mM NaCl, 1 mM $MnCl_2$) over 30 min at 37 °C.

### Surface preparation
In this, 96-well glass plates (Matek, PBK96G-1.5-5-F) were incubated with 80 μl of 100 μg/ml BSA-Biotin (Thermo Fisher, 29130) in PBS for 2 h at room temperature. In specified contexts (such as with the SNS), 18.75 μg/ml of fibronectin was added to the BSA-biotin solution. Wells were rinsed 2x with cold PBS and incubated with 100 μg/ml neutravidin (Thermo Fisher, 31000) for 30 min at room temperature. Wells were rinsed once more and incubated with 80 μl of 1 μM RAD-TGT and were incubated at 4 °C overnight. It is essential that the wells never dry to prevent any unintentional adsorption to the surface; for all experiments volume in the well never fell below 50 μl.

### Cell lines
U251 glioma cells were a gift from the Odde lab, and CHO-K1 cells were purchased from ATCC (ATCC, CCL-61). U251 talinKO and U251 CD44 KO cells were prepared using published methods[60,61]. Briefly, TLN1 KO and CD44 KO were achieved using the CRISPR/Cas9 system. A guide RNA (sequence AACUGUGAAGACGAUCAUGG) was created to target TLN1. A guide RNA (gRNA) was created to target exon 2 in CD44 human cell lines (GAATACACCTGCAAAGCGGC). A co-transposition method was used to enhance screening for knockout clones. Briefly, cells were transfected with Cas9 nuclease, gRNA, PiggyBac transposases, and PiggyBac transposon plasmid containing puromycin selection using FuGENE (Promega, Madison, WI) following the manufacturer's protocol. Puromycin selection was performed, and single-cell clones were generated using serial dilution. After transfections, the cells were split into single clones, and western blot was used to confirm the knockout. WB also verified that there was no overexpression of Talin2 in response to KO of Talin1.

### Cell culture
Cells were maintained for no more than 15 passages. All cells were regularly passaged every 2–3 days when 80% confluency was achieved. All cells, other than CHO-K1 cells, were maintained in Dulbecco's Modified Eagle Medium (Corning, 10027CV) supplemented with 10% fetal bovine serum (FBS) (R&D Systems, S11150) and penicillin/streptomycin (Gibco,15070063). CHO-K1 cells were maintained in an F-12K medium (ATCC, 302004) supplemented with 10% FBS and penicillin/streptomycin.

## RAD-TGT experimental setup

Cells of interest were trypsinized (Gibco, 25200056) for 5 min and transferred to Opti-MEM Reduced Serum Media (Gibco, 31985062). Following this, cells were counted using an automated cell counter (Countess II, Invitrogen). If cells were to be treated with para-amino-Blebbistatin (Cayman Chemical, 22699), the cells were incubated with 50 μM para-amino-Blebbistatin at 37 °C for 30 min before application to the RAD-TGT surface. All para-amino-Blebbistatin experiments were accompanied by a vehicle control consisting of cells being incubated in an equivalent volume of DMSO (Invitrogen, D12345). RAD-TGT surfaces were washed once with cold PBS and twice with Opti-MEM. Following washes and any drug treatment, 15,000 cells were added to the wells and incubated at 37 °C with 5% $CO_2$ for the desired period of time, typically 90 min.

## Flow cytometry experiments

Following incubation, the RAD-TGT medium was removed, and 30 μl of trypsin was added to each well. The plate was further incubated at 37 °C for 5 min. Following trypsinization, 170 μl of flow cytometry buffer (PBS with 1% FBS and 1 mM EDTA) was added to the wells to quench the trypsin and resuspend the cells. Cells were removed from wells and, without further washing, analyzed with a BD Accuri C6 Plus Personal Flow Cytometer using the BD Accuri C6 Plus (64-bit) Software Version 1.0.34.1. Cells were gated from the collected data, followed by gating for individual cells and then gating for cells that had a detectable signal (Supplementary Fig. 18).

## Sequencing experiments

Sequencing experiments followed the same method as flow cytometry experiments, but cells were not analyzed with a cytometer. Three unique RAD-TGTs were in the experimental well, one for each ligand (WDV, FN, or Echi), and each RAD-TGT contained a novel barcode on the 3′ end of the ligand strand. Following removal from the well, cells were treated with 1x Passive Lysis Buffer (Promega, E1941) for 20 min at room temperature while on an orbital shaker. The lysate had the barcoded top strands PCR amplified, and the resulting PCR product was gel purified and submitted for Sanger and Illumina Sequencing. Quantification of barcodes in the Sanger sequences was performed using the Base-editing analysis software EditR[62]. A dummy guide sequence was denoted that spanned the barcode area. Percentages at AGA and GAG in the barcodes were averaged to calculate the percentage of Echi and FN, respectively, for forward and reverse sequencings of three biological replicates. WDV was calculated from the difference of 100%. The program Tracy[63] was used to visualize the sequence displayed in Fig. 3. Quantification of barcodes in the next-generation sequencing assay was performed with a custom Python script using the Biopython package[64]. Briefly, forward sequencing reads were parsed, trimmed to include only the region containing the barcode, and barcodes were then counted. Reverse sequencing reads were reverse complemented and then processed in an identical manner. Forward and reverse counts for the three barcodes in each sample were combined, and barcodes corresponding to either Echi, FN, or WDV were exported for each sample.

## Microscopy

To visualize TGT ruptures, surfaces were prepared following the above protocols. TGTs consisted of Quencher Ligand Strand and a Fluorescently Labeled Anchor Strand, so if ruptured, one should observe a gain of fluorescence on the glass surface. Rather than dissociating the cells post-trypsinization, each well had the media removed, and Fluorobrite DMEM (Gibco, A1896701) was immediately added to the wells. Cells were then imaged at ×40 with an EVOS FL AUTO fluorescent microscope. Images were processed with ImageJ. To create the outline of cells in qTGT images, the accompanying brightfield images had their perimeter traced in ImageJ. The traced perimeters were saved as regions of interest (ROI), and the equivalent ROI was identified in the fluorescent images. The superimposed ROI was then converted into a dotted line using the Dotted Line plug-in.

## Statistics and reproducibility

In general, each experiment was performed on at least three separate days with 1–3 separate wells per day. The SuperPlots were generated by importing histogram data from FlowJo into Prism. The ligand data were normalized to the median of the WDV alone data. The fluorescence data were concatenated to produce the dot plots. The medians of the normalized data for each biological replicate were used to overlay on the dot plot, and statistical significance was calculated using a one-way ANOVA or paired two-tailed $t$-test as previously described[40]. The sample size was dependent on cell confluency in a 96-well plate; no statistical method was used to predetermine the sample size. No data were excluded except during flow cytometry gating, which gated out cell debris, dead cells, and cell clusters. Furthermore, events with 0 fluorescence were excluded; this appeared to be an artifact from the cytometer; even in cases of no fluorophore present, cells had an intrinsic fluorescent value and, on random dates, also had some 0-value events. We concluded, due to the cells' intrinsic fluorescence value that the 0 values were not cells. All control conditions (WDV only) and accompanying experimental conditions (WDV-Echi or WDV-FN) utilized cells that were cultured together and only separated at the time of plating. No further randomization was performed on experiments, and investigators were not blinded to allocation during experiments and outcome assessment. All images shown are representative of outcomes from at least three biological replicates.

## Reporting summary

Further information on research design is available in the Nature Portfolio Reporting Summary linked to this article.

## Data availability

The raw data from barcoding experiments is provided with the code in the "Code Availability" section. Raw sequencing data is also available from the Sequence Read Archive under the BioProject accession number "PRJNA956930" (SRA accession numbers "SRX20001005", "SRX20001006", and "SRX20001007"). Source data are provided with this paper.

## Code availability

The code generated and used in this report can be found at https://github.com/AdamTSmiley/TGT_Barcode_Counter.

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

## Acknowledgements

W.R.G. acknowledges funding from the NIH R35GM119483 and U54 CA210190. W.R.G. is a Pew Biomedical Scholar. M.D.K. acknowledges a Cancer Center Training grant (T32CA009138). D.J.O. acknowledges funding from NIH grants: U54 CA210190, P01 CA254849, and U54 CA268069.

## Author contributions

M.R.P. performed the majority of the experiments in this study, analyzed data, and wrote the manuscript. A.T.S. performed Sanger and NGS sequencing experiments. M.P.R. analyzed imaging data. M.D.K. performed experiments related to this study. W.R.G. analyzed data and wrote the manuscript. G.A.S., S.M.A., B.A.S., D.A.L., and D.J.O. engineered the CRISPR-KO cell lines used in this study.

## Competing interests

M.R.P. and W.R.G. are inventors on U.S. Provisional Patent application no. 63/359,612 regarding the assay developed in this work. The Regents of the University of Minnesota is the assignee of this patent application. The remaining authors report no competing interests.
