## [Peer Review File · Nature Communications]

Reviewers' comments:

Reviewer #1 (Remarks to the Author):

This paper presents a flow cytometry-based assay, RAD-TGT, to evaluate cell biomechanical ability in terms of exerting contractile force by integrins. The central component of the assay is a dye-labeled DNA duplex immobilized on surfaces. The DNA is also labeled by echistatin (a broad-spectrum integrin ligand) via HUH-endonuclease tag (An awesome technique developed in the authors' lab). Cells plated on these DNA-immobilized surfaces can pluck off the dye-labeled DNA strand, which is either clinging to the cell surface or somehow internalized to cells. Either way, the cells are eventually fluorescent and can be read by a flow cytometry. The principle was demonstrated with two cell-lines and several variants of DNA constructs. Cells with talin 1 or CD44 KO were also tested to demonstrate that the result of RAD-TGT assay is correlated with the KO of the two proteins. The authors also demonstrated using PCR to read target DNA amount in cell lysate which reports cell force without the need of fluorescence.

Overall, the principle of the assay was well explained and the feasibility was shown. However, this work has some major limitations:

1. Potential of this assay was not convincingly demonstrated. Flow cytometry can certainly sample thousands to millions of cells, with each cell as a single data point. However, under what circumstance this high throughput is needed? Without proper justification, the millions of cells could just be a histogram of fluorescence levels in cells, where hundreds of cells versus millions of cells may not make a big difference. Although the authors showed dual-color flow cytometry of RAD-TGT versus talin KO in cells. The new assay may not be better than conventional immunostaining. The authors claimed "The high throughput readout of RAD-TGT rupture could enable rapid screening for knockout of cellular proteins that alter cellular force generation." However, this indirect approach does not seem to be better than a simple immunostaining of the target proteins in KO cells (with control) and subsequent flow cytometry or cell imaging.

2. The core principles of using flow cytometry and DNA sequencing to report cell force have been demonstrated in "DNA-based microparticle tension sensors (μ TS) for measuring cell mechanics in non-planar geometries and for high-throughput quantification", *Angewandte Chemie*, 60, 2-9, and "Mechanically-triggered Hybridization Chain Reaction", *Angewandte Chemie*, 60, 19974-19981. The former paper uses tension probe-coated microbeads to test cells and go through flow cytometry, and the latter one uses rolling circle replication to report force generated by cells on the substrate. To be fair, the authors referenced both works and the RAD-TGT assay is not the same as the two works, but it shares similarity in terms of the working principles.

3. There is a major issue related to whether the fluorescence in cells is really and only associated with cell force. First, it is not clear where the fluorescence comes from. Is the dye-DNA present on cell membrane or internalized to cells? There will be issues for both cases. If former, the integrin-ligand affinity, instead of integrin-mediated force, could be a determinant factor of the fluorescence level in RAD-TGT assay, and cells may harvest soluble dye-DNA (due to DNA spontaneous dissociation) in the solution by directly binding to integrin ligands. If latter, how dye-DNA enters cells is not clear (Might be related to integrin recycling?). The fluorescence level in cells is likely not only determined by force that pulls off DNAs, but also determined by the efficiency of ligand internalization, degradation, etc. The nature and origina of fluroescence in cells should be examined and discussed.

Minor Issue:

“qTGT” is not defined in the manuscript. “12pN and 54 pN TGTs” are referenced, but a brief explanation in the context would be helpful.

In sFig. 10, the labels in first figure is confusing. What is “FN DMSO no FN”?

Reviewer #2 (Remarks to the Author):

The paper by Pawlak and colleagues shows a clever approach to use flow cytometry to approximate the traction forces generated by cells. The authors use the TGT probes which are double stranded DNA sequences that dissociate due to mechanical forces applied by cell surface receptors. Rather than detect the signal accumulating on the surface using microscopy (as is the norm in the field), the team places a fluorophore on the top DNA strand and detects the accumulation of signal in/on the cell. This approach to read out should increase throughput. Other claims of novelty in this brief manuscript focus on the use of an engineered endonuclease to link target proteins to the nucleic acid as well as the ability to use commercial polystyrene or glass slides for anchoring the TGT to the surface.

The paper shows a broad range of experiments and controls in the main text as well as a number of controls in the SI. The data is rather weak and lacks rigor and it seems that the manuscript was hastily assembled. In some cases the data in the SI contradicts the data in the main text and this reviewer was not convinced that the RAD probes behave as described. The role of non-specific binding and release of RAD probes is likely significant. Also, the influence of nucleases in the release of RAD probes is well documented and not addressed. The precedent in the field with two published reports showing approaches to increasing the throughput of the TGT using flow and 96well plates further diminish the novelty of the work. Accordingly, I am not supportive of publication of the work in its current form. I

suspect that the authors could bring the work up to the quality of a paper in Nature Communications after running additional experiments and controls and addressing gaps and deficiencies outlined below.

1. There are actually two reports that describe published work to bypass the need for high resolution imaging for TGT readout. One report was correctly cited (as ref. 24) and describes the use of flow cytometry to increase throughput. However, the current manuscript failed to cite work showing the use of plate readers for 96-well plate readout of TGT signal which was published in *Angew Chem Int. Ed.* in 2016 (55(18), 5488-5492). Given these two papers, the novelty of using flow to readout the TGT signal is rather limited.

2. Prior work with the TGT already reported that the ruptured TGT is internalized into cells. Please see this paper: *Biophys J.* 2015 Dec 1; 109(11): 2259–2267 which has a section titled “dsDNA in TGT is separated into single strands and internalized”. There is a detailed figure analyzing the internalized top strand TGT. Therefore, the fundamental mechanism of RAD-TGT is already well known and documented in the literature but not appropriately credited here.

3. Another inaccurate claim in the paper is that “Current TGTs also require specialized glass surfaces and arduous chemical modifications of DNA to conjugate TGTs to ligands/surface, limiting their potential for high throughput assays and widespread use.” Published work has already addressed these problems. For example, this paper showed a convenient approach to modify 96-well plates with TGT using biotinylated-BSA (*J Vis Exp.* 2019 Apr 25; (146): 10.3791/59476). In *Scientific Reports* volume 6, Article number: 21584 (2016), Ha and colleagues developed a Protein G (ProG)-based force sensor in which force-reporting tethers are conjugated to ProG instead of ligands. Here a recombinant ligand fused with IgG-Fc is conveniently assembled with the force sensor through ProG:Fc binding, therefore avoiding ligand conjugation and purification processes. This approach has been used in a number of subsequent reports.

4. One fundamental flaw in using the RAD-TGT approach to quantify cell forces is that the probability of probe internalization following rupture is cell line and cell state dependent. The dissociation of ligand from its receptor following rupture is plausible but this is not discussed. SI Figure 8 begins to address this point but the data is not corrected using this information. Can one use RAD-TGT to compare two different cell types or the same cells from different passages or cell cycles?

5. The microscopy data shown in figure 1d shows two images but these are very confusing as no brightfield or DIC imaging is included to identify where the cells are and even how many cells are present. The reader is left wondering what is actually shown in the black box that is the control with WDV only. In fact this is a major weakness of all the presented data. There is dearth of microscopy

validation to confirm that the TGT probes are rupturing under the cells and the associated uptake of probes within the cell.

6. The claim that “We observed that U251 cells exhibit an enhanced fold change in median fluorescence compared to CHO-K1 cells (Fig 2a,b), suggesting they exert higher cellular forces.” is inaccurate. The greater signal may be due to larger cell size or a greater number of surface displayed integrins. It could also be due to more frequent forces and more rapid turnover rate of integrin. It could also be a greater k_{on}/k_{off} rate. In fact, I see no evidence that the forces are greater in magnitude. That requires a different set of measurements.

7. SI Fig. 3 is not very useful as the control group of cells incubated on WDV surfaces is missing. Moreover, experimental details are missing and it isn't clear if this is the 12 or 56 pN probes.

8. The data in figure 2C is weak as the QY and brightness of each dye is different and the differing intensities may not necessarily mean very much. I would suggest swapping the fluorescent tags to validate the result or using calibration standards so that the relative uptake of the 12 and 56 pN probe is measured.

9. SI figure 7 contradicts the main data figures. The group labeled WDV-Echi+Blebb shows identical signal to the positive group lacking Blebb. This is a major concern as it shows that the RAD signal is simply due to the ultra-high affinity of the Echi ligand that binds receptors regardless of myosin contractility.

10. The FN ligand is more physiological but unfortunately the signal generated by the FN is weak and seems noisy. SI Fig. 9 further confirms this weakness of the work. Key microscopy analysis of why the FN ligand only shows a minor increase in signal is needed.

11. SI Fig. 11 shows no difference between WDV negative control and FN. This contradicts the main text figures and highlights a major flaw in the work that suggests that the RAD signal may be due to binding of non-specifically adhered ligand rather than mechanical rupture as claimed.

12. The gating strategy (SI Fig. 12) where non-fluorescent cells are eliminated and the lack of a live-dead stain is concerning here. Further controls are needed to demonstrate that this gating strategy is quantitative as it can eliminate a significant number of data points that would bias the “fold change” metric used throughout the work.

13. It is very strange that CD44 knockout led to similar levels of dampening of tension as that of talin knockout. The brightfield images in SI Fig. 6 are consistent with the literature – showing a drastic change in cell morphology upon knockout of talin. The minor or perhaps non-detectable change in cell morphology following CD44 knockout is also consistent with literature. Talin is central to formation of focal adhesions while CD44 is dispensable. However the tension data shown in the main figures is inconsistent with SI Figure 6 as it shown statistically identical RAD signal for CD44 and talin KO cells (Fig2e). How can that be? Further analysis and controls are needed to address this contradiction.

14. SI Figure 2 shows identical RAD flow signal for 12 and 56 pN probes at all time points tested from 10 min to 120 with the exception of the 120 min data point that showed marginally greater signal. This data is troubling as it contradicts almost 9 years of work with the TGT that shows that the rupture of the 12 pN probes is always more significant than that of the 56 pN probe. The data just doesn't make sense and needs further explanation and likely further analysis to rule out the possibility that the data is due to an artifact. The lack of microscopy data makes it very difficult to understand why the data is contradictory to the past literature.

15. Prior work investigating the release of nucleases by cells and their activity on TGTs (J. Biophotonics. 2019; 12:e201800351.) clearly shows that nucleases will target TGTs within the time window investigated here. There are no experiments or controls addressing the potential role of nucleases leading to false positive RAD-TGT signal. This must be addressed and weaken the rigor of the work.

16. The sequencing experiments shown in figure 3 are very preliminary and lack appropriate controls that confirm the claims.

Reviewer #3 (Remarks to the Author):

Single molecule forces can control cellular signaling mediated by membrane receptors, and examples include T cell receptors, Notch receptors and integrins. Therefore, tools to quantify such forces in situ are valuable. Pawiak et al makes a nice set of contributions to the toolset by showing the rupture of tension gauge tethers (TGTs) can be read out in high throughput using flow cytometry and using DNA sequencing through barcoding of TGTs. These are interesting and compelling proof of principle

experiments and I would recommend publication of a suitably revised manuscript in Nature Communications.

1. Let me start with a relatively minor point. I do not believe that a new acronym (RAD-TGT) is necessary. Internalization (or endocytosis) of fluorescently labeled, liganded DNA strand was already reported previously (see Figure 4 of Wang et al, Biophysical Journal (PMID 26636937). Reading out receptor-specific mechanical history through measuring internalized DNA is not specific to their ligand tethering strategy, whether flow cytometry or barcode sequencing is used. I agree that their ligand conjugation strategy can be advantageous for some applications but it does not enable anything that the existing strategies do not already.

2. Probably the most important point. Single molecule force sensors are powerful because they can measure forces at the single molecule level. In the case of integrins, they can define forces that are required to activate signaling through single integrin-ligand bonds. What's lacking in the current version of the manuscript is a demonstration that their new readouts can provide information on single molecular forces. When they show that different ligands, inhibitors and gene knockdowns can change flow cytometry signals, they cannot tell, at least currently, whether the changes come from changes in the magnitude of single molecule forces through single integrins or changes from the number of single-integrin ligand bonds.

3. A related point to point #2 above is that the new readouts need to be validated against conventional TGT readouts. For example, TGTs do not allow cell adhesion on TGTs weaker than 40 pN (Wang and Ha, Science 2013), and for the new readouts to be useful, they need to be able to recapitulate at least some of the true single molecule force determination.

4. Another related point is that the new readouts that rely on TGT internalization may not be sensitive enough for practical applications. Most of the data come from a protein that binds all integrins with high affinity and when they used a fibronectin derived ligand, the signal was greatly reduced to the degree that it is unlikely that they would be able to detect the effect of inhibitors and gene knockdown. A possible and probable explanation is that the protein ligand gives the most favorable situation where all of the integrins in the cell contribute to the signal, and the stronger signal is mainly due to the large of number of integrins engaged instead of coming from stronger single molecular forces. This is an important point because their readouts may not be useful for more typical applications where only a subset of integrins is targeted.

5. I am not sure what controls have been to show that the signal measured in flow cytometry is due to internalized TGTs instead of TGTs bound to the cell surface.

6. Their ligand conjugation approach, an elegant method the Gordon lab developed previously, requires a single stranded overhang. As nuclease activities would be more severe against single stranded DNA compared to double stranded DNA, I am wondering if this is causing any issues. They do mention the use of serum free medium to avoid DNA degradation by nucleases. See also Pal et al, JCB 2021 PMID 33904858.

7. The effect of blebbistatin additional is surprisingly modest. An earlier study by Wang et al (referred in #2 above) showed almost completed inhibition of TGT rupture by myosin II inhibition. Perhaps the modest effect is due to the use of the protein ligand that binds all integrins?

8. In discussing Fig. 2ab, the authors suggest that U251 exerts higher cellular forces than CHO-K1. It is unclear what they mean. There is more TGT internalized but it does not mean that each TGT experiences a stronger force. It could be that U251 has more integrins.

9. Page 5. "Generally, the 54pN RAD-TGTs resulted in similar trends as the 12pN tension tolerance." This is surprising because from CHO-K1 studies in Wang et al 2013 and 2015 (referred to above), rupture behavior was completely different between the 12 pN and 54 pN TGTs.

10. Page 6. "Interestingly, we observed a statistically significant increase in HUH-echistatin relative to HUH and HUH-FN with both sequencing methods but we only saw a statistically significant increase in HUH-FN with sanger sequencing." -"increase in HUH-FN" should be changed to "increase in HUH-FN relative HUH"

11. Figure 2 color schemes may be confusing to some readers. Orange and cyan colors are used for different things in panel a vs panel c.

Figure S12. Para-amino is spelled out for one panel but not for the other.

Thanks to all the reviewers for very helpful comments. Our manuscript is greatly improved because of
the additional experiments and additional clarifications/explanations added to the manuscript. We have
provided a point by point response to all the reviewers below, but wanted to summarize the major
changes we made in response to concerns shared by more than one reviewer.

**Major experiments added:**

- 1. We performed the majority of experiments using both 12 and 54 pN TGTs (the previous version
focused on 12pN TGTs). In this submission, both tension tolerance TGTs contained a donor
fluorophore (ruptured strand) and quencher (anchor strand) to eliminate signal from detachment
of intact duplexes (Fig1, Fig2, Fig3, Fig S1). These new experiments prompted by reviewers
provided additional insights and, for instance, revealed that knockout of CD44 does indeed have
a different and more muted effect on cell mechanics than knockout of talin.
- 2. We provide more comprehensive validation of RAD-TGTs to ensure RAD-TGTs recapitulate prior
literature:
 - - Adhesion of CHO-K1 cells on TGTs of different tension tolerances and ligands
 - - qTGT “turn on” fluorescence imaging of CHO-K1 cells on TGTs of different tension
tolerances and ligands
 - - Blebbistatin treatment of CHO-K1 cells
 - - Table of relevant TGT studies and how our results correspond to previous findings
- 3. We performed additional experiments on mixed populations of cells using 12pN TGTs or
multiplexed 12/54pN TGTs to demonstrate that RAD-TGTs can distinguish populations of cells
with distinct mechanotypes. (Fig 3)

**Major points addressed/clarified in the manuscript**

- 1. Clarified why we aimed to develop a high throughput assay using DNA tension probes; our assay
takes advantage of recording the mechanical history of a cell via rupture of TGTs but instead **IN**
**THE CELL of interest**. This importantly means that distinct populations of cells that differentially
exert tension on TGTs, or have a different “mechanotype”, can be mixed and measured in the
same well by the level of fluorescence in the cell via flow cytometry. This will be useful for high
throughput CRISPR and drug screening of agents that alter mechanotype.
- 2. Further discussion of the inter-related factors that contribute to the observed fluorescence signal;
such as adhesion, affinity of ligand, integrin availability and engagement, traction forces, and
spread area
- 3. Reorganized manuscript; Figure 1 is now all CHO-K1 benchmarking of our flow cytometry
readout and echistatin ligand against traditional surface imaging. Figure 2 demonstrates the use
of RAD-TGTs in U251 glioma cells with both flow cytometry and NGS readouts. Figure 3 focuses
on CRISPR-KO cell lines as proof of concept for future CRISPR-KO screens.
- 4. **Addressing “contradictory” data in former SI Fig7- now SI Fig8**. Several reviewers were
concerned that SI figure 7 contradicted the data in our study. We would like to clarify that **this is**
**not the case** and we have more clearly referred to the figure in the text and in the figure legend.
These data are presented to demonstrate why we chose to work with glass bottomed plates in all
of the assays in the study because polystyrene plates yield unsatisfactory results (this is what is
shown in the figure); broad fluorescence in contrast to narrow distributions.

45 **Reviewer 1**

46
47 *(comments) This paper presents a flow cytometry-based assay, RAD-TGT, to evaluate cell*
48 *biomechanical ability in terms of exerting contractile force by integrins. The central component of the*

1 assay is a dye-labeled DNA duplex immobilized on surfaces. The DNA is also labeled by echistatin (a
2 broad-spectrum integrin ligand) via HUH-endonuclease tag (An awesome technique developed in the
3 authors' lab). Cells plated on these DNA-immobilized surfaces can pluck off the dye-labeled DNA strand,
which is either clinging to the cell surface or somehow internalized to cells. Either way, the cells are
eventually fluorescent and can be read by a flow cytometry. The principle was demonstrated with two
cell-lines and several variants of DNA constructs. Cells with talin 1 or CD44 KO were also tested to
demonstrate that the result of RAD-TGT assay is correlated with the KO of the two proteins. The authors
also demonstrated using PCR to read target DNA amount in cell lysate which reports cell force without
the need of fluorescence. Overall, the principle of the assay was well explained and the feasibility was
shown. However, this work has some major limitations:

**1-1.** Potential of this assay was not convincingly demonstrated. Flow cytometry can certainly sample
thousands to millions of cells, with each cell as a single data point. However, under what circumstance
this high throughput is needed? Without proper justification, the millions of cells could just be a histogram
of fluorescence levels in cells, where hundreds of cells versus millions of cells may not make a big
difference. Although the authors showed dual-color flow cytometry of RAD-TGT versus talin KO in cells.
The new assay may not be better than conventional immunostaining. The authors claimed "The high
throughput readout of RAD-TGT rupture could enable rapid screening for knockout of cellular proteins
that alter cellular force generation." However, this indirect approach does not seem to be better than a
simple immunostaining of the target proteins in KO cells (with control) and subsequent flow cytometry or
cell imaging.

Thank you for this comment- we have attempted to clarify the need for a high throughput assay to
measure a cell's mechanical phenotype, or mechanotype. As with all TGT assays, our RAD-TGT assay
records the mechanical history of the cell via rupture of immobilized DNA duplex tension probes.
However, unlike traditional surface imaging, our assay measures uptake of ruptured DNA tension probes
**in the cell of interest by flow cytometry.** Since diseased cells exhibit distinct mechanotypes compared
to normal cells, our assay will permit high-throughput screens such as CRISPR knockout or drug screens
to identify genes or drugs that alter cellular mechanotype. Genome-wide CRISPR KO screens require on
the order of 20 million cells for rigor; this would not be possible with immunostaining and reading out
intensities of tens of cells using imaging. The flow cytometry based readout will allow pooled CRISPR
screens where one can flow sort cells with higher or lower fluorescence to identify genes that determine
a cellular mechanotype. The DNA sequencing-based readout can be used for drug screens or arrayed
CRISPR screens.

1-2. The core principles of using flow cytometry and DNA sequencing to report cell force have been demonstrated in "DNA-based microparticle tension sensors (μ TS) for measuring cell mechanics in non-planar geometries and for high-throughput quantification", *Angewandte Chemie*, 60, 2-9, and "Mechanically-triggered Hybridization Chain Reaction", *Angewandte Chemie*, 60, 19974-19981. The former paper uses tension probecoated microbeads to test cells and go through flow cytometry, and the latter one uses rolling circle replication to report force generated by cells on the substrate. To be fair, the

1 authors referenced both works and the RADTGT assay is not the same as the two works, but it shares
similarity in terms of the working principles.

We expanded the discussion of these assays in our manuscript. We highlighted that the main advance
of our assay is that it records the mechanical history of the cell **in the cell of interest**. The main
drawback of the DNA tension probes immobilized on beads or of amplification of the “scar” DNA
remaining on the surface after duplex rupture is that they cannot be used to detect differences in force
between cells with different mechanotypes in mixed populations of cells since the sensor is on a bead or
surface. Moreover, the study that used bead based DNA tension probes with a flow cytometry readout
(Hu, et al. Angew. Chem., 2021) showed quite modest changes in fluorescence signal between the
sample and negative control - the bead population broadened but did not shift in median fluorescence as
in our assay.

**1-3.** There is a major issue related to whether the fluorescence in cells is really and only associated with
cell force. First, it is not clear where the fluorescence comes from. Is the dye-DNA present on cell
membrane or internalized to cells? There will be issues for both cases. If former, the integrin-ligand
affinity, instead of integrin-mediated force, could be a determinant factor of the fluorescence level in
RAD-TGT assay, and cells may harvest soluble dye-DNA (due to DNA spontaneous dissociation) in the
solution by directly binding to integrin ligands. If latter, how dye-DNA enters cells is not clear (Might be
related to integrin recycling?). The fluorescence level in cells is likely not only determined by force that
pulls off DNAs, but also determined by the efficiency of ligand internalization, degradation, etc. The
nature and origin of fluorescence in cells should be examined and discussed.

To address the issue of whether the fluorescence is outside the cell or internalized, we performed an
experiment commonly used to assess delivery of DNA origami structures. Here, the fluorescence signal
is compared with and without DNase treatment; DNase can degrade only the DNA that may be stuck to
the outside of the cell. These experiments indicated that a very small percentage of our signal may be
due to oligos remaining on the surface that have not yet been internalized by integrin recycling. We
would further argue that whether the DNA is outside of the cell or internalized, it would still reflect
ruptured DNA duplexes and be read out equivalently by flow cytometry given our pH independent Cy5
dye.

Regarding the high affinity of the echistatin for integrins- indeed we expect that the higher affinity
(together with the broader spectrum of integrins bound) contribute to the higher signal we see compared
to fibronectin-based ligands. Since our goal is to record the cumulative mechanical history after a
specified time and compare this relative signal to cells in another condition, we view this higher signal as
a perk of the assay.

**1-4.** Minor Issue:

“qTGT” is not defined in the manuscript. “12pN and 54 pN TGTs” are referenced, but a brief explanation
in the context would be helpful.

We have now defined this and referenced relevant papers. We have also noted the use of Integrated
Tension Sensor “ITS” to describe some DNA duplex-based tension sensors.

**1-5.** In sFig. 10, the labels in first figure is confusing. What is “FN DMSO no FN”?

We have clarified this on the Figure- this means fibronectin domain conjugated to TGT but no additional
fibronectin coated on plate

**Reviewer 2**

*The paper by Pawlak and colleagues shows a clever approach to use flow cytometry to approximate the*
*traction forces generated by cells. The authors use the TGT probes which are double stranded DNA*
*sequences that dissociate due to mechanical forces applied by cell surface receptors. Rather than detect*
*the signal accumulating on the surface using microscopy (as is the norm in the field), the team places a*
*fluorophore on the top DNA strand and detects the accumulation of signal in/on the cell. This approach to*
*read out should increase throughput. Other claims of novelty in this brief manuscript focus on the use of*
*an engineered endonuclease to link target proteins to the nucleic acid as well as the ability to use*
*commercial polystyrene or glass slides for anchoring the TGT to the surface.*

Thank you for recognizing the highlights of this approach.

*The paper shows a broad range of experiments and controls in the main text as well as a number of*
*controls in the SI. The data is rather weak and lacks rigor and it seems that the manuscript was hastily*
*assembled. In some cases the data in the SI contradicts the data in the main text and this reviewer was*
*not convinced that the RAD probes behave as described. The role of non-specific binding and release of*
*RAD probes is likely significant. Also, the influence of nucleases in the release of RAD probes is well*
*documented and not addressed. The precedent in the field with two published reports showing*
*approaches to increasing the throughput of the TGT using flow and 96well plates further diminish the*
*novelty of the work.*

*Accordingly, I am not supportive of publication of the work in its current form. I suspect that the authors*
*could bring the work up to the quality of a paper in Nature Communications after running additional*
*experiments and controls and addressing gaps and deficiencies outlined below.*

These points addressed within comments below

**2-1.** There are actually two reports that describe published work to bypass the need for high resolution
imaging for TGT readout. One report was correctly cited (as ref. 24) and describes the use of flow
cytometry to increase throughput. However, the current manuscript failed to cite work showing the use of
plate readers for 96-well plate readout of TGT signal which was published in Angew Chem Int. Ed. in
2016 (55(18), 5488-5492). Given these two papers, the novelty of using flow to readout the TGT signal is
rather limited.

As explained in Response 1-2 above, the novelty of our work is that our assay records the mechanical
history of the cell via uptake of ruptured DNA tension probes in the cell of interest. But we agree that
the other two assays needed further explanation in our manuscript!

**2-2.** Prior work with the TGT already reported that the ruptured TGT is internalized into cells. Please see
this paper: Biophys J. 2015 Dec 1; 109(11): 2259–2267 which has a section titled “dsDNA in TGT is
separated into single strands and internalized”. There is a detailed figure analyzing the internalized top
strand TGT.

Therefore, the fundamental mechanism of RAD-TGT is already well known and documented in the
literature but not appropriately credited here.

Thank you for pointing this out- we have noted this study and properly credited this work. This work was
indeed part of the premise for applying the TGT assay to read out internalized fluorescence using flow
cytometry.

**2-3.** Another inaccurate claim in the paper is that “Current TGTs also require specialized glass surfaces
and arduous chemical modifications of DNA to conjugate TGTs to ligands/surface, limiting their potential
for high throughput assays and widespread use.” Published work has already addressed these problems.
For example, this paper showed a convenient approach to modify 96-well plates with TGT using
biotinylated-BSA (J Vis Exp. 2019 Apr 25; (146): 10.3791/59476). In Scientific Reports volume 6, Article
number: 21584 (2016), Ha and colleagues developed a Protein G (ProG)-based force sensor in which
force-reporting tethers are conjugated to ProG instead of ligands. Here a recombinant ligand fused with
IgG-Fc is conveniently assembled with the force sensor through ProG:Fc binding, therefore avoiding
ligand conjugation and purification processes. This approach has been used in a number of subsequent
reports.

We have toned down the language around biotin-BSA coating of the commercial glass plates, as it is
much more common than we realized. We did track surface coating and type of plate for studies cited in
Supplementary Table 1. We have also cited the proteinG study, and pointed out that the advance of this
method is that HUH-tags allow direct and covalent attachment of ligands/antibodies without steps such
as the chemical linking of ProteinG to the TGT.

**2-4.** One fundamental flaw in using the RAD-TGT approach to quantify cell forces is that the probability of
probe internalization following rupture is cell line and cell state dependent. The dissociation of ligand
from its receptor following rupture is plausible but this is not discussed. SI Figure 8 begins to address this
point but the data is not corrected using this information. Can one use RAD-TGT to compare two
different cell types or the same cells from different passages or cell cycles?

Yes! Just as different cell types and cell states exhibit different traction forces when measured by
traction force microscopy, we believe our data demonstrates that different cell types rupture and
internalize TGTs differentially due to different properties of the cell, such as integrin composition,
contractility, adhesivity, cell size.

**2-5.** The microscopy data shown in figure 1d shows two images but these are very confusing as no
brightfield or DIC imaging is included to identify where the cells are and even how many cells are
present. The reader is left wondering what is actually shown in the black box that is the control with WDV
only. In fact this is a major weakness of all the presented data. There is dearth of microscopy validation
to confirm that the TGT probes are rupturing under the cells and the associated uptake of probes within
the cell.

We have performed both brightfield imaging and complementary qTGT assays to benchmark RAD-TGTs
with the TGTs used by others as shown in Figure 1. We have also included a table in the supplement
comparing several of our observations with previous studies. Below is a list of what our imaging and
benchmarking demonstrates in the context of the main findings of prior TGT work.

Findings in TGT literature:

- 1. CHO-K1 cells do not adhere to 12pN TGTs conjugated to cyclic-RGD (cRGD) ligands, but do
adhere and spread on 54pN TGTs. **Our data on TGTs conjugated to a fibronectin domain**
**recapitulate this (Fig 1b).**
- 2. Gain of (or loss of) fluorescence patterns under cells plated on 12pN TGTs is bright and constant
under whole cell footprint. Fluorescence pattern under cells plated on 54pN TGTs occurs in
streaks likely consistent with forces exerted by focal adhesions. **Our data on TGTs conjugated to**
**a fibronectin domain recapitulate this (Fig 1c).**
- 3. There are two quantification parameters presented for the gain/loss of fluorescence of ruptured
TGTs in the literature (see supplementary table 1). The **ratio of TGTs ruptured** under a cell
footprint and the **total integrated fluorescence** under a given cell footprint. It should be noted
that **our flow readout should be comparable to the total integrated fluorescence**. Prior
studies show that the ratio of ruptured TGTs is HIGHER for 12pN TGTs than for 54pN TGTs,
since the whole cell area shows fluorescence for 12pN in contrast to the streaks in the case of
54pN TGTs. However, whether the integrated fluorescence under a given cell footprint is higher
for 12pN vs 54pN TGTs **depends on the size of the cell** and thus its adherence properties on
the cRGD TGTs. As mentioned above and shown in Fig 1b, CHO-K1 cells do not adhere well to
cRGD 12pN TGTs. When no cell adhesion enhancers are added to the experiment, the cells on
12pN surfaces have very small footprints and the integrated intensity under the cell can be lower
than 54pN. Though the fluorescence is weaker for 54pN, the area is much larger. However,
many TGT studies add fibronectin or poly-lysine PEG to TGTs to facilitate adhesion, or plate cells
on mixed tension tolerance TGTs; in these cases, cells are equally well spread on 12 and 45 pN
TGTs, and thus the 12pN has higher integrated fluorescence. **Our data on TGTs conjugated to a**
**fibronectin domain recapitulate this. On 12pN fibronectin TGTs, the median fluorescence is lower**
**for 12pN than 54pN. However, on echistatin ligands where cells are well spread for both 12 and**
**54pN TGTs, the 12pN shows higher median fluorescence than 54pN.**
- 4. Blebbistatin: previous studies show variable effects on CHO-K1 rupture of 12pN TGTs and more
drastic effects on 54pN TGTs (see Supplementary Table 1). Our data on echistatin-conjugated
TGTs show about 60% reduction in signal for both 12 and 54pN TGTs, likely to the broader
spectrum of integrins targeted by echistatin.

Other conclusions from the imaging

- 1. CHO-K1 cells plated on echistatin ligands are well spread out in both 12 and 54pN TGTs. This is
likely a combination of binding to other integrins that cyclic RGD cannot bind well to, which may
exert forces between 12 and 54 pN as well as sub-nanomolar affinity of the ligand for all RGD
binding ligands.
- 2. Cells do not adhere well to the negative control WDV conjugated TGTs. Moreover, there is no
fluorescent signal on qTGTs plated on WDV-TGTs. These suggest there is not non-specific
binding of cells to our RAD-TGTs. Moreover, these data demonstrate that surface nucleases are
not an issue
(see also Supp Fig 4B where we test for surface nuclease activity using a different configuration
of qTGT)
- 3. Fold change in CY5 fluorescence is much higher for echistatin than fibronectin

**2-6** The claim that “We observed that U251 cells exhibit an enhanced fold change in median
fluorescence compared to CHO-K1 cells (Fig 2a,b), suggesting they exert higher cellular forces.” is
inaccurate. The greater signal may be due to larger cell size or a greater number of surface displayed
integrins. It could also be due to more frequent forces and more rapid turnover rate of integrin. It could
also be a greater kon/koff rate. In fact, I see no evidence that the forces are greater in magnitude. That
requires a different set of measurements.

This is exactly the goal of our work- to use TGTs to provide relative differences in mechanotype of mixed populations of cells because their cell specific properties have led to differential interactions with TGTs. As the reviewer points out, the mechanical history resulting from uptake of ruptured TGTs will depend on many factors, such as affinity of ligand for receptor, types of integrins on the cell surface in the case of echistatin ligands, contractility, adhesiveness. Current SI Figure 13 shows a titration of soluble echistatin for two different cell types, and demonstrates a higher affinity of echistatin for U251 cells than CHO-K1 cells. Since number of integrins correlates with traction force generated of cells according to the motor clutch model, we concluded that U251 cells exert more cellular force on RAD-TGTs than CHO-K1 cells.

We would also like to point out that all of our data is normalized against WDV along surfaces to increase the rigor of our results. While other TGT studies usually show a TGT lacking ligand to demonstrate specificity, the Rupture ratios and/or integrated fluorescence reported is not corrected for the no ligand condition like ours.

2-7 SI Fig. 3 is not very useful as the control group of cells incubated on WDV surfaces is missing. Moreover, experimental details are missing and it isn't clear if this is the 12 or 56 pN probes.

This is now SI Fig 6 and we have clarified in legend

2-8 The data in figure 2C is weak as the QY and brightness of each dye is different and the differing intensities may not necessarily mean very much. I would suggest swapping the fluorescent tags to validate the result or using calibration standards so that the relative uptake of the 12 and 56 pN probe is measured.

Our first submission did not involve many experiments with 54pN probes. Thanks to the Reviewers, we realized this was a deficiency of the study, and have performed many more experiments with both 12 and 54 pN RAD-TGTs. Moreover, we utilized tension probes that included a quencher on the bottom strand so that any aberrant removal of intact DNA duplexes from the surface from cell handling would not contribute to our signal. Our first submission did not use a 54pN TGT with the quencher.

For the multiplexed 12 and 54pN probes, we performed the experiment with Cy5 and Alexa488, and interchanged them for comparison. Figure 3e shows a 12pN Cy5/54pN Alexa488 experiment while the reverse is supplementary Fig 16. There are subtle differences but overall the profiles are similar.

2-9 SI figure 7 contradicts the main data figures. The group labeled WDV-Echi+Blebb shows identical signal to the positive group lacking Blebb. This is a major concern as it shows that the RAD signal is simply due to the ultra-high affinity of the Echi ligand that binds receptors regardless of myosin contractility.

We apologize that this was not more clear- the point of SI Fig 7 (now SI Fig 8) was to demonstrate why we did NOT use polystyrene plates in these studies! Exactly as you observed, on polystyrene plates, the fluorescence histograms were not clean peaks like they are on glass, making it difficult to discern differences under different conditions. We used glass for all of the experiments in the paper. We changed the figure legend and clarified this point in the main text.

**2-10** The FN ligand is more physiological but unfortunately the signal generated by the FN is weak and
seems noisy. SI Fig. 9 further confirms this weakness of the work. Key microscopy analysis of why the
FN ligand only shows a minor increase in signal is needed.

In Figure 1, we now show both brightfield and qTGT experiments on WDV alone, fibronectin, and
echistatin. We also compare these results to all relevant literature TGT studies and find that our work
recapitulates previous observations.

Figure 1 brightfield images show:

- 1) WDV alone: Very little cell adhesion.
- 2) Fibronectin-WDV: Very little cell adhesion on 12pN TGTs, and well spread cells on 54pN TGTs.
This has been observed multiple times for CHO-K1 cells in the literature (see supplementary
Table 1). It should be noted that many TGT studies in the literature add cell adhesion agents to
improve behavior of cells on 12pN TGT surfaces, such as fibronectin or poly-lysine PEG.
- 3) Echistatin-WDV: Well spread cells on both 12 and 54pN TGTs.

Figure 1 qTGT fluorescence images show:

- 1) WDV alone: no fluorescence. This also speaks to absence of nuclease activity, though we
designed a separate set of TGTs to test this (See Response 2-15)
- 2) Fibronectin-WDV: 12pN TGT bright uniform fluorescence under cells with very small footprint
(due to lack of adhesion); 54pN TGT- streaks of fluorescence corresponding to focal adhesion,
though over a much larger cell area.
- 3) Echistatin-WDV: Fluorescence under similar cell footprints for 12pN/54pN; fluorescence brighter
under 12pN

Figure 1 flow cytometry RAD-TGT assay shows:

- 1) WDV alone: background median fluorescence 850
- 2) Fibronectin-WDV: 12pN Median fluorescence about 1055, 54 pN median fluorescence 1100
- 3) Echistatin-WDV: 12pN MFI 6000, 54 pN MFI ~3000.

**2-11** SI Fig. 11 shows no difference between WDV negative control and FN. This contradicts the main
text figures and highlights a major flaw in the work that suggests that the RAD signal may be due to
binding of nonspecifically adhered ligand rather than mechanical rupture as claimed.

For the DNA barcode experiments, the TGTs containing WDV, Fibronectin-WDV, and Echistatin-WDV
were all plated in one well at $\frac{1}{3}$ the concentration compared to other experiments. Thus it does not
contradict other data- it simply decreases the already modest fold-change for the FN ligand below the
background. We have made this more clear in results and Figure legend

**2-12** The gating strategy (SI Fig. 12) where non-fluorescent cells are eliminated and the lack of a live-
dead stain is concerning here. Further controls are need to demonstrate that this gating strategy is
quantitative as it can eliminate a significant number of data points that would bias the “fold change”
metric used throughout the work.

The BD Accuri benchtop flow cytometer used does not allow any voltages to be changed. We also did
not set a threshold and this is likely electronic noise and not actual cell or even cell debris. We did
perform live dead staining which did not label this signal. Moreover, this population appears even with
negative control plated cells on no TGTs. This is common cell profile using the Accuri BD for many cell
types.

**2-13** It is very strange that CD44 knockout led to similar levels of dampening of tension as that of talin
knockout. The brightfield images in SI Fig. 6 are consistent with the literature – showing a drastic change
in cell morphology upon knockout of talin. The minor or perhaps non-detectable change in cell
morphology following CD44 knockout is also consistent with literature. Talin is central to formation of
focal adhesions while CD44 is dispensable. However the tension data shown in the main figures is
inconsistent with SI Figure 6 as it shown statistically identical RAD signal for CD44 and talin KO cells
(Fig2e). How can that be? Further analysis and controls are needed to address this contradiction.

Thank you for pointing this out. There is much evidence in the literature for crosstalk between cd44 and
integrins that would explain its knockout effecting overall force (See Refs 51 and 52 in main text).
However, the reviewer comment prompted us to repeat these experiments with both 12 and 54pN TGTs
including the quencher on the bottom strand to remove spurious rupture from the data. As shown in the
previous submission, knockout of both talin and CD44 in U251 cells shows 25-35% decrease in signal for
the 12pN TGT. However, when we looked at these cell lines on 54pN TGTs, we observe distinct rupture
behavior for the two cell lines. The talinKO showed a similar decrease in signal as on the 12pN TGT, but
the CD44-KO did not significantly decrease the 54pN TGT signal. This suggests mechanistic differences
in how these knockouts alter cellular forces. As expected, talin appears to affect both actomyosin-
independent and dependent forces, while the CD44 KO does not affect actomyosin dependent forces
which have been demonstrated to be above 40pN of force.

**2-14** SI Figure 2 shows identical RAD flow signal for 12 and 56 pN probes at all time points tested from
10 min to 120min, with the exception of the 120 min data point that showed marginally greater signal.
This data is troubling as it contradicts almost 9 years of work with the TGT that shows that the rupture of
the 12 pN probes is always more significant than that of the 56 pN probe. The data just doesn't make
sense and needs further explanation and likely further analysis to rule out the possibility that the data is
due to an artifact. The lack of microscopy data makes it very difficult to understand why the data is
contradictory to the past literature.

In our previous work, we had mostly focused on the 12pN TGT, for which we had employed a quencher
on the bottom strand to correct for any intact duplexes that ruptures. New Fig S1 shows the effect of the
quencher strand on the flow data. In that work, we did not have the quencher version of the 54pN.
However, it should also be noted that most TGT works report % ruptured under a cell which is different
from this readout, which is more akin to integrated fluorescence under a cell.

**2-15** Prior work investigating the release of nucleases by cells and their activity on TGTs (J.
Biophotonics. 2019; 12:e201800351.) clearly shows that nucleases will target TGTS within the time
window investigated here. There are no experiments or controls addressing the potential role of
nucleases leading to false positive RAD-TGT signal. This must be addressed and weaken the rigor of the
work.

We took care to use serum free media to mitigate nuclease effects. As the reviewer points out, there are
reports in the literature of nuclease activity by surface proteases. Our qTGT imaging in Figure 1 shows
no fluorescence well for cells plated on WDV-TGTs alone. Moreover, Supplementary Fig 11 uses
another qTGT derivative known as a surface nuclease sensor (developed by the Wang Group) which is
essentially a qTGT without any ligand present so that only nuclease activity may lead to duplex
dissociation and gain of fluorescence. These studies reveal minimal fluorescence indicating minimal
nuclease activity. In our flow experiments, we always normalize to the WDV alone control which would
remove effects of nuclease activity from our data.

2-16. The sequencing experiments shown in figure 3 are very preliminary and lack appropriate controls that confirm the claims.

We agree that the sequencing experiment proved to be more of a confirmation of other experiments in the paper with a novel readout, but not a novel experiment warranting its own Figure. Thus we grouped sequencing results with corresponding flow experiments of U251 cells on WDV, fibronectin, and echistatin ligands in Figure 2

Reviewer 3

Single molecule forces can control cellular signaling mediated by membrane receptors, and examples include T cell receptors, Notch receptors and integrins. Therefore, tools to quantify such forces in situ are valuable.

Pawiak et al makes a nice set of contributions to the toolset by showing the rupture of tension gauge tethers (TGTs) can be read out in high throughput using flow cytometry and using DNA sequencing through barcoding of TGTs. These are interesting and compelling proof of principle experiments and I would recommend publication of a suitably revised manuscript in Nature Communications.

3-1 Let me start with a relatively minor point. I do not believe that a new acronym (RAD-TGT) is necessary.

Internalization (or endocytosis) of fluorescently labeled, liganded DNA strand was already reported previously (see Figure 4 of Wang et al, Biophysical Journal (PMID 26636937). Reading out receptor-specific mechanical history through measuring internalized DNA is not specific to their ligand tethering strategy, whether flow cytometry or barcode sequencing is used. I agree that their ligand conjugation strategy can be advantageous for some applications but it does not enable anything that the existing strategies do not already.

We appreciate this sentiment; we wanted to be able to refer to these in short form but staying true to the original TGT platform for which they are based. There are many derivative names of TGTs in the literature such as quenched-TGTs (qTGTs) and multiplexed-TGTs (mTGTs) as well as new names such as integrated tension sensor (ITS).

3-2 Probably the most important point. Single molecule force sensors are powerful because they can measure forces at the single molecule level. In the case of integrins, they can define forces that are required to activate signaling through single integrin-ligand bonds. What's lacking in the current version of the manuscript is a demonstration that their new readouts can provide information on single molecular forces. When they show that different ligands, inhibitors and gene knockdowns can change flow cytometry signals, they cannot tell, at least currently, whether the changes come from changes in the magnitude of single molecule forces through single integrins or changes from the number of single-integrin ligand bonds.

We agree that this may appear to be a deficit of the study, but our goal was to move away from gleaning
information about single molecule forces and take advantage of the ability these DNA tension probes to
provide a recording of cumulative TGT rupture at a given time that can be used to provide a rapid
reading of a parameter of cellular mechanotype.

**3-3A** related point to point #2 above is that the new readouts need to be validated against conventional
TGT readouts. For example, TGTs do not allow cell adhesion on TGTs weaker than 40 pN (Wang and
Ha, Science 2013), and for the new readouts to be useful, they need to be able to recapitulate at least
some of the true single molecule force determination.

We have performed multiple new experiments as well as compiled other TGT results to compare to our
results. Namely we have performed brightfield and qTGT imaging in Figure 1 to compare to our flow
measurements on WDV, fibronectin-WDV, and echistatin-WDV.

**3-4** Another related point is that the new readouts that rely on TGT internalization may not be sensitive
enough for practical applications. Most of the data come from a protein that binds all integrins with high
affinity and when they used a fibronectin derived ligand, the signal was greatly reduced to the degree
that it is unlikely that they would be able to detect the effect of inhibitors and gene knockdown. A possible
and probable explanation is that the protein ligand gives the most favorable situation where all of the
integrins in the cell contribute to the signal, and the stronger signal is mainly due to the large number
of integrins engaged instead of coming from stronger single molecular forces. This is an important point
because their readouts may not be useful for more typical applications where only a subset of integrins is
targeted.

We would argue that this assay would be useful as a kind of high throughput traction force microscopy,
and that you would want to use echistatin and other high affinity ligands targeting other integrins or even
high affinity antibodies to probe mechanotypes of interest. Indeed, as explained above, more integrins
being engaged likely leads to higher signal which is useful in comparing multiple cell types or to identify
genes or drugs in CRISPR-KO or drug screens that alter that mechanotype signature.

**3-5** I am not sure what controls have been to show that the signal measured in flow cytometry is due to
internalized TGTs instead of TGTs bound to the cell surface.

See Response 1-3 above.

**3-6b** Their ligand conjugation approach, an elegant method the Gordon lab developed previously,
requires a single stranded overhang. As nuclease activities would be more severe against single
stranded DNA compared to double stranded DNA, I am wondering if this is causing any issues. They do
mention the use of serum free medium to avoid DNA degradation by nucleases. See also Pal et al, JCB
2021 PMID 33904858.

See Response 2-15 above.

**3-7** The effect of blebbistatin additional is surprisingly modest. An earlier study by Wang et al (referred in
#2 above) showed almost completed inhibition of TGT rupture by myosin II inhibition. Perhaps the
modest effect is due to the use of the protein ligand that binds all integrins?

In our compilation of TGT studies, the effect of blebbistatin varied a lot depending on the cell type, the
sensitivity of the readout, and the definition of "signal" being ratio of ruptured under a cell area versus
integrated fluorescence under a cell area. Again, most early TGT measurements focused on changes to
the rupture ratio which we are not measuring and the effects on tens of individual cells. Studies that
measure integrated fluorescence showed blebbistatin changes of 10 to 60% similar to our
measurements. Moreover, since echistatin is probing integrins that have not been well studied, it is
possible that our assay samples additional integrins with activities independent of actomyosin activity.

**3-8** In discussing Fig. 2ab, the authors suggest that U251 exerts higher cellular forces than CHO-K1. It is
unclear what they mean. There is more TGT internalized but it does not mean that each TGT
experiences a stronger force. It could be that U251 has more integrins.

We have tried to make this more clear in the introduction and result. Indeed we do not mean that
individual TGTs are experiencing more force but that more TGTs are ruptured (in part due to more
integrins and higher affinity ligand).

**3-9**Page 5. "Generally, the 54pN RAD-TGTs resulted in similar trends as the 12pN tension tolerance."
This is surprising because from CHO-K1 studies in Wang et al 2013 and 2015 (referred to above),
rupture behavior was completely different between the 12 pN and 54 pN TGTs.

Thank you for pointing this out. First of all, our first submission did not use a 54pN probe with a
quencher to correct for spurious rupture of duplexes. Our results when both 12 and 54 pN probes are
quenched unless the duplex is melted shows more fluorescence in cells with 12pN probes, consistent
with our qTGT studies in Fig 1 and with the literature. Though again, some TGT studies report rupture
ratio which is different from integrated intensity, and often gives different results if cells are not well
spread on both 12 and 54 pN probes.

**3-10**Page 6. "Interestingly, we observed a statistically significant increase in HUH-echistatin relative to
HUH and HUH-FN with both sequencing methods but we only saw a statistically significant increase in
HUH-FN with sanger sequencing." -"increase in HUH-FN" should be changed to "increase in HUH-FN
relative HUH"

Thank you for pointing this out

**3-11**Figure 2 color schemes may be confusing to some readers. Orange and cyan colors are used for
different things in panel a vs panel c.

Figure S12. Para-amino is spelled out for one panel but not for the other.

We have made colors in figures more consistent and made para-amino consistent.

REVIEWER COMMENTS

Reviewer #1 (Remarks to the Author):

I appreciate the authors' revisions. Here, I merely provide my point of view for the editor's deliberation. The two major concerns raised up in the last round of revision have not been well addressed. These two concerns are:

1. Justification for high throughput screening using RAD-TGT.

This work demonstrated well that RAD-TGT based flow cytometry is possible, but even after revision, it does not demonstrate the advantage of RAD-TGT based flow cytometry over the old-fashioned immunostaining or direct force imaging using fluorescent tension probes. All the assays in this paper, such as force alteration by talin KO, CD44 KO and blebbistatin treatment, can also be examined by direct force imaging. No direct evidence shows that the high throughput is necessary. Since the authors claim that RAD-TGT based flow cytometry can be integrated with pooled CRISPR OR -omics screening, a proof-of-concept test can be done by treating cells with Genome-Scale CRISPR Knock-Out (GeCKO) libraries which are commercially available (not expensive), and performing RAD-TGT based flow cytometry afterwards.

2. Rigor for the interpretation of RAD-TGT fluorescence signal in cells.

Because RAD-TGT is an indirect approach for cell mechanotype examination, there are multiple potential cellular processes contributed to RAD-TGT signals in cells, other than integrin forces, likely also including cell endocytosis efficiency, extracellular substance degradation efficiency in cells. In this regard, using RAD-TGT signals to report cell mechanotype alone could suffer from significant false positive read-outs caused by non-force related sources. While authors showed that RAD-TGT fluorescence is altered in cells when putative mechanosensing proteins are knocked out, it's highly possible that the RAD-TGT fluorescence would also be altered in cells when endocytic proteins or proteins related to degradation of extracellular substances are disrupted. Put in short, this paper did a good job proving that cell mechanotype alteration leads to RAD-TGT signal change, but have no evidence showing that RAD-TGT signal in cells is solely determined by cell mechanotype.

Reviewer #2 (Remarks to the Author):

Peer review

The current paper does indeed represent an improvement over the past submission with the addition of new data and discussion. I had to treat this as a new submission – since the vast majority of data and even some of the conclusions have been updated since the first submission. The authors now give credit to past work that demonstrated internalization of ruptured DNA and also credit to past work that developed flow cytometry based-analysis of TGT rupture. I still remain supportive of the novelty of the work and particularly with the early proof-of-concept experiments to conduct sequencing on internalized probes and also to knockout talin and CD44 and show that the probes generate unique signals for each cell population. That said, the work still needs major revisions to increase rigor. The primary issue is that some of the claims are not fully supported by the data and there are claims that are likely incorrect or perhaps inconsistent across figures. Rigor is weak in several SI figures and just because experiments are shown in the SI does not mean that replicates and statistical testing are no longer needed. Below are the issues organized by importance (major versus minor).

Major issue: line 249. The statement that “Remarkably, the quantification of ruptured TGTs by sequencing corresponds very well to the flow cytometry results.” Is not accurate. The sequencing analysis showed a difference between WDV and FN in sanger analysis but not by NGS. The flow data showed no difference between WDV and FN. This may be in disagreement with figure 2a but I can't tell because there is no statistical test of that data. Also the WDV group analyzed by sanger in figure 2d seems to show that all three replicates had identical means which is peculiar. Finally the caption of figure 2 needs to explain what the data points and error bars mean.

Major issue: The data shown with talinKO is surprising and is inconsistent with the literature as it is well known that talin is essential for focal adhesion formation. The SI figure confirms this and shows rounded cells with talinKO. Hence, I am surprised to see strong flow signal in talinKO cells (Figure 3a) for both 12 and 54 pN. I am most surprised by the 54pN data. Ha and others have already shown that rupture of 54 pN duplexes requires the full engagement of focal adhesions. The authors should show fluorescence images to validate that talinKO generate strong DNA duplex rupture events. This approach to validation was used in Figure 1 and 2 and should be included for figure 3 as well.

Major concern: There is no statistical test of the claim that the 12pN-WDV-Echi signal is weaker than the 56pN-WDV-Echi signal in Figure 1. This analysis should be included to support this claim.

Major concern: SI Figure 12: The nuclease sensor data is not displayed correctly. The activity of nucleases would result in a negative signal (from a probe that lacks quencher). However the representative images are contrasted to show the background as zero (black). Any loss of signal due to nucleases would simply not be observable in the way the data is currently shown. The signal under the cell must be quantitatively compared against the signal outside the cell. Please adjust contrast. Moreover, to support the claim properly, there needs to be some quantitation and showing a representative image is insufficient.

Major concern: Statistical testing is absent from figure 2a and 2b. Also, the main text discusses the Echi probes in figure 2 a,b but ignores the FN data. Please include a discussion of that data. I am concerned that the FN data shows contrasting response when compared to the Echi data. At least this is what it appears from the data. Why would that be the case? It does not make sense to me and suggests other mechanisms are contributing to the signal.

Major point: The statement in line 143 that “Nevertheless, these results indicate that the assembled TGTs function as expected and that surface preparations are equivalent to prior studies.” Needs to be backed up quantitatively. This includes measuring the cell spreading area from representative BF images in Fig. 1b and also quantifying the fluorescence images in Figure 1c.

Major point: The claim that “higher signal for 12 pN than 54pN TGTs” is based on representative figure 1c and SI figure 3 and this needs quantitation. The claims can not stand on their feet without some type of quantitation.

Major point: Supplementary Figure 1: The plots indicate that the signal for the 54 pN probe was greater than that for the 12 pN probe for the no-quencher probes. This is contradictory to the data shown in main figure 1 c and d. The figure caption does not indicate how the Cy5 intensity was measured. Please clarify. Also the caption needs to describe what the horizontal lines in the violin plots mean (CI, SEM...etc?).

Major point: The representative flow histograms in figure 1D and fluorescence data in SI figure 3 are not consistent. Flow shows similar signal for WDV-FN 12 and 56 pN while microscopy shows very different signal. This is in contrast to WDV-Echi where flow and imaging seem to qualitatively match. Why would the microscopy and flow agree in one case but not in the other?

Major concern: SI Fig 4B, SI Figure 5, SI Figure 6, and SI Figure 7 seem to be the result of single measurements and are not replicated and thus lacks statistical rigor. I may be mistaken but there is no error bars and no description of statistics in the captions or plots. SI figure 9 shows duplicates of

experiments and in some cases triplicates but the caption fails to explain what colors and data points mean.

Major concern: line 226 states “As expected, rupture of both 12 and 54pN TGTs was decreased upon treatment para-amino-Blebbistatin”. However SI Figure 11 is not consistent with this claim and the 54pN TGT did not show statistically significant decrease in signal following Bleb treatment.

Minor issue: Line 236 states “We observed U251 cells bound 3-4 fold more than CHO-K1 cells, suggesting a greater number of integrin clutches available to exert traction force.” I think this conclusion is not fully correct. The greater bound probe on U251 cells may be the result of many other factors such as cell size, copy number of integrins displayed on the membrane, and also the competition between FN binding and Echi binding for each cell types.

Minor point: Figure 1b: It is surprising to see that CHO-K1 cells cultured on the 12pN TGT probes to spread when the ligand is Echi. This finding strongly suggests that the Echi ligand may irreversibly bind to integrins and also it may partially activate integrin receptors in the absence of tension. This response is cell line dependent as U251 cells showed lower cell spreading on 12pN Echi surfaces compared to that of the 54pN-Echi surfaces. The discussion needs to address this confusing point as it is an inherent quirk of using Echi as a ligand.

Minor point: Fig. 1C: the accompanying brightfield images are absent from main figure and shown in SI. These are needed so that the readers can compare the location of the fluorescence signal with the location of the cell. It is very difficult to make sense of the data without knowing where cells are located and the overlap of the tension signal to the cell location. Another issue is that the fluorescence contrasts in SI figure 3 is different from that of figure 1c.

Minor point: Descriptions such as “Streaks” and “puncta” around line 150 are vague and need better descriptions.

Minor point: The conclusion of SI Figure 6 is highly dependent on the size of area of the substrate so this information should be included.

Minor typo: Pg 3. Line 118: change alpha5 to alphaV.

Reviewer #3 (Remarks to the Author):

The manuscript has been improved significantly with additional experiments and clarifications. I now understand that the main application of the technology will be in high throughput screen of mechanotypes using the most favorable ligand. The authors also present an early demonstration of 'screening based on mechanotypes' here using a mixture of cells and mixture of TGTs. I recommend publication.

1. I noticed that WDV is never defined, and sometimes used in lower case.
2. The authors performed experiments to support the idea that cell fluorescence comes from internalized TGTs. The most direct demonstration would be to show that there is strong TGT fluorescence inside the cells based on microscopy data. It is curious why they do not present such data.

Reviewer #1 (Remarks to the Author):

I appreciate the authors' revisions. Here, I merely provide my point of view for the editor's deliberation. The two major concerns raised up in the last round of revision have not been well addressed. These two concerns are:

1. Justification for high throughput screening using RAD-TGT.

This work demonstrated well that RAD-TGT based flow cytometry is possible, but even after revision, it does not demonstrate the advantage of RAD-TGT based flow cytometry over the old-fashioned immunostaining or direct force imaging using fluorescent tension probes. All the assays in this paper, such as force alteration by talin KO, CD44 KO and blebbistatin treatment, can also be examined by direct force imaging. No direct evidence shows that the high throughput is necessary. Since the authors claim that RAD-TGT based flow cytometry can be integrated with pooled CRISPR OR -omics screening, a proof-of-concept test can be done by treating cells with Genome-Scale CRISPR Knock-Out (GeCKO) libraries which are commercially available (not expensive), and performing RAD-TGT based flow cytometry afterwards.

- While it is true that traditional methods can measure differences in force exerted on DNA tension probes by knockout cell lines and drug treatments, our method provides the first **proof of concept** that *the cell of interest can be fluorescently-tagged in proportion with cumulative TGT rupture*, to allow sorting and further analysis of cells with an observed mechanotype. We feel this is a novel application of DNA tension probes for which proof of concept warrants publication. **Traditional imaging-based TGT studies cannot distinguish cells** according to their mechanotype in a mixed populations of cells. We **added several experiments in Figure 3** to demonstrate that flow cytometry can be used to distinguish populations of cells with different mechanical histories or mechanotypes, which will inspire many new lines of research such as CRISPR screens. We are indeed performing a CRISPR screen in our lab and using our assay to sort according to mechanotype. Yes, libraries are commercially available and available in Addgene. But we have already spent months constitutively expressing Cas9 in our cells of interest, working out lentivirus titering, etc. and thus this type of experiment is beyond the scope of this manuscript.
- Moreover, the flow cytometry readout is further beneficial as is often difficult to perform intensity quantification from fluorescence images, as intensities can be affected by surface imperfections, light settings, background differences, photobleaching, etc. Researchers typically characterize dozens of cells, while we can characterize thousands in an unbiased way.

2. Rigor for the interpretation of RAD-TGT fluorescence signal in cells.

Because RAD-TGT is an indirect approach for cell mechanotype examination, there are multiple potential cellular processes contributed to RAD-TGT signals in cells, other than integrin forces, likely also including cell endocytosis efficiency, extracellular substance degradation efficiency in cells. In this regard, using RAD-TGT signals to report cell mechanotype alone could suffer from significant false positive read-outs caused by non-force related sources. While authors showed that RAD-TGT fluorescence is altered in cells when putative mechanosensing proteins are knocked out, it's highly possible that the RAD-TGT fluorescence would also be altered in cells when endocytic proteins or proteins related to degradation of extracellular substances are disrupted. Put in short, this paper did a good job proving that cell mechanotype alteration leads to RAD-TGT signal change, but have no evidence showing that RAD-TGT signal in cells is solely determined by cell mechanotype.

- We totally agree that factors other than forces transmitted across integrins contribute to the mechanotype signal our assay is reporting. Though, we would contend that most of these factors are intertwined with integrin-mediated forces, such as integrin surface expression and degradation, cell spread area, and endocytic trafficking. It should be noted that traditional TGT studies often report the integrated fluorescence under a cell as "cumulative fluorescence", which is also subject to the same caveats that signal can depend on factors other than integrin forces. Regarding differential endocytic rates, though endocytosis may not be directly involved in force transmission across integrins, it is still an actin mediated process that is likely involved in crosstalk with integrin mediated forces. In addition, our cell would be tagged with the fluorescent TGT whether the oligo was stuck on the outside or internalized (Supplementary Fig 4B), so differences in internalization rates should not play a significant role in differences between mechanotype. We did attempt to explain this in the first revision, and we apologize it was not sufficient. We have further attempted to

emphasize that our readout likely depends on many factors, and is a measure of relative mechanotype between different types or treatments of cells. We now have this paragraph in the discussion:

- "Here we demonstrate that RAD-TGTs report significant differences in signal following changes in cellular generated forces. However, other factors outside of direct force application through integrins may alter signal, such as cell spread area, variability in receptor expression/degradation, affinity for ligand, efficiency of endocytosis, and ECM remodeling between cells. These factors likely all contribute to observed cellular mechanotypes. Moreover, the fluorescence could also be influenced by factors unrelated to cellular forces, such as cell size. Thus caution should be taken in interpreting differences in fluorescence signal between different cell types. Nevertheless, RAD-TGTs offer a rapid screen to identify putative mechanotype modulators which can be further validated with mechanistic studies."

Reviewer #2 (Remarks to the Author):

Peer review

The current paper does indeed represent an improvement over the past submission with the addition of new data and discussion. I had to treat this as a new submission – since the vast majority of data and even some of the conclusions have been updated since the first submission. The authors now give credit to past work that demonstrated internalization of ruptured DNA and also credit to past work that developed flow cytometry based-analysis of TGT rupture. **I still remain supportive of the novelty of the work and particularly with the early proof-of-concept experiments to conduct sequencing on internalized probes and also to knockout talin and CD44 and show that the probes generate unique signals for each cell population.** That said, the work still needs major revisions to increase rigor. The primary issue is that some of the claims are not fully supported by the data and there are claims that are likely incorrect or perhaps inconsistent across figures. Rigor is weak in several SI figures and just because experiments are shown in the SI does not mean that replicates and statistical testing are no longer needed. Below are the issues organized by importance (major versus minor).

Major issue: line 249. The statement that "Remarkably, the quantification of ruptured TGTs by sequencing corresponds very well to the flow cytometry results." Is not accurate. The sequencing analysis showed a difference between WDV and FN in sanger analysis but not by NGS. The flow data showed no difference between WDV and FN. This may be in disagreement with figure 2a but I can't tell because there is no statistical test of that data. Also the WDV group analyzed by sanger in figure 2d seems to show that all three replicates had identical means which is peculiar. Finally the caption of figure 2 needs to explain what the data points and error bars mean.

- We apologize for not including the statistics in Figure 2a and the details of statistics in the legend. We have now added these. For these data, we have also (Supplementary Fig 10A) compared two different methods for analyzing biological replicates of flow cytometry histograms; 1) comparing the medians of cell populations, as we did in all our our experiments, and 2) comparing %positive cells, which is often used in the literature but is less stringent.
- Regarding the subtle differences between sequencing by Sanger and NGS- to us it is remarkable that they are so similar as they are completely different readouts and analyses! It is known that Sanger is often more sensitive than NGS for this type of analysis ¹. The WDV Sanger read (as described in methods) was calculated for each sequencing read by adding the FN and Echi peak percentages and subtracting from 100. The sum of the FN and Echi peak percentages reads was very consistent, explaining why the WDV reads are so close (but not exact- the medians of the reads for the three biological replicates are 17.2, 17.45, and 17.4). Clearly, at 1/3 ligand concentration used in the sequencing experiments, the signal from FN-WDV is near the limit of detection of the assay.
- Finally- the experiments in Figure 2a and 2d/supplementary Fig 14 are different in that only 1 ligand was present per well in 2a while 2d and supp. 14 had all 3 ligands present. Thus the concentration of a given ligand is cut by 1/3 in Fig 2d/Supp14. Thus they cannot be directly compared with each other. We further clarified this lower ligand concentration in the Figure legends.
- It seems the Reviewer is questioning the rigor of our studies. We would like to emphasize that we attempted to analyze our data in the most rigorous way possible. Many cell based studies simply perform statistics on all the cells in a given population in comparison to another treatment containing a population of many cells. This results in artificially low p-values and masks experimental variability. Other studies using flow cytometry

referred to in our paper (e.g. ²) simply set a gate and count percent positive cells. Statistical significance is calculated between %positive values derived from several biological replicates. Indeed we explored this option and found the statistical significance to be much more generous using %positive than using the median of the population. We observed the median fluorescence which reflects every cell in the population to be the most stringent measure, and decided to move forward with this analysis. We have added a Supplementary Figure 10A to illustrate this point- comparing the percent positive used by others in the field to the median fluorescence method for the data in Fig 2A.

Major issue: The data shown with talinKO is surprising and is inconsistent with the literature as it is well known that talin is essential for focal adhesion formation. The SI figure confirms this and shows rounded cells with talinKO. Hence, I am surprised to see strong flow signal in talinKO cells (Figure 3a) for both 12 and 54 pN. I am most surprised by the 54pN data. Ha and others have already shown that rupture of 54 pN duplexes requires the full engagement of focal adhesions. The authors should show fluorescence images to validate that talinKO generate strong DNA duplex rupture events. This approach to validation was used in Figure 1 and 2 and should be included for figure 3 as well.

- Complete loss of talin would be expected to ablate all force, but talin exists in two forms, talin 1 and talin 2. Within the text we mention that talinKO is in reference to talin 1 ko only, recent reports have demonstrated that both talin 1 and talin 2 are capable of forming and maintaining focal adhesions following loss of the other isoform ³, this study also confirmed a drastic decrease in cell spreading much like images we demonstrate.

Major concern: There is no statistical test of the claim that the 12pN-WDV-Echi signal is weaker than the 56pN-WDV-Echi signal in Figure 1. This analysis should be included to support this claim.

- For echistatin, the 12pN signal is HIGHER than the 54pN, as is expected when cells adhere well. For FN, where the cells adhere poorly to 12pN TGTs, the 12pN signal is slightly weaker. This is due to the small spread area. We have added quantification of cumulative fluorescence intensity for the cells in Fig 1c in Supplementary Figure 3

Major concern: SI Figure 12: The nuclease sensor data is not displayed correctly. The activity of nucleases would result in a negative signal (from a probe that lacks quencher). However the representative images are contrasted to show the background as zero (black). Any loss of signal due to nucleases would simply not be observable in the way the data is currently shown. The signal under the cell must be quantitatively compared against the signal outside the cell. Please adjust contrast. Moreover, to support the claim properly, there needs to be some quantitation and showing a representative image is insufficient.

- Our description of the Surface Nuclease Sensor used was insufficient in the last revision; we adapted nuclease sensors previously used ⁴ containing a fluorophore on the bottom strand proximal to the biotin modification and quencher directly above on the top strand so that following degradation, the remaining fixed oligo fragment gains fluorescence. We have added cartoons to SI Figure 12.

Major concern: Statistical testing is absent from figure 2a and 2b. Also, the main text discusses the Echi probes in figure 2 a,b but ignores the FN data. Please include a discussion of that data. I am concerned that the FN data shows contrasting response when compared to the Echi data. At least this is what it appears from the data. Why would that be the case? It does not make sense to me and suggests other mechanisms are contributing to the signal.

- We have added statistical testing for Fig2A. We have expanded our discussion about differences observed between ligands, which we believe are due to Echistatin being able to bind more integrin subtypes with much greater affinity than FN. This is supported by recent published work (cited within the main text) demonstrating different integrin subtypes mediate biological events in different force regimes. Furthermore we highlight this by demonstrating that U251 cells readily adhere to 12 pN Echi surfaces and struggle to adhere to the 12 pN FN surface as seen in Main Figure 1 and supplementary figures 10 and 15

Major point: The statement in line 143 that “Nevertheless, these results indicate that the assembled TGTs function as expected and that surface preparations are equivalent to prior studies.” Needs to be backed up quantitatively. This includes measuring the cell spreading area from representative BF images in Fig. 1b and also quantifying the fluorescence images in Figure 1c.

- The gold standard TGT studies simply show brightfield images of cells on 12 and 54 pN TGTs and fluorescent images quantifying % TGT rupture under a cell footprint. Cell areas are difficult to measure when cells are not adhering. We feel the adhesion deficits on WDV and 12pN FN-WDV TGTs are obvious in Figure 1. However, we have provided cell area measurements for relevant drug treatments and cell line comparisons (Supplemental Fig 9). We have quantified fluorescence intensity density per cell in Supplementary Figure 3.

Major point: The claim that “higher signal for 12 pN than 54pN TGTs” is based on representative figure 1c and SI figure 3 and this needs quantitation. The claims can not stand on their feet without some type of quantitation.

- SEE ABOVE

Major point: Supplementary Figure 1: The plots indicate that the signal for the 54 pN probe was greater than that for the 12 pN probe for the no-quencher probes. This is contradictory to the data shown in main figure 1 c and d. The figure caption does not indicate how the Cy5 intensity was measured. Please clarify. Also the caption needs to describe what the horizontal lines in the violin plots mean (CI, SEM...etc?).

- We would like to emphasize that the point of supp fig 1 is to demonstrate that the quencher is necessary to mitigate any signal that may occur if the entire RAD-TGT binds to/enters a cell. This may come from mechanical perturbation during handling, trypsin mediated degradation of BSA and subsequent release of entire TGT, or excess unbound TGT in solution. All of the experiments in the manuscript use the TGT containing the quencher, this figure is to demonstrate WHY we added the quencher.

Major point: The representative flow histograms in figure 1D and fluorescence data in SI figure 3 are not consistent. Flow shows similar signal for WDV-FN 12 and 56 pN while microscopy shows very different signal. This is in contrast to WDV-Echi where flow and imaging seem to qualitatively match. Why would the microscopy and flow agree in one case but not in the other?

- Cells do not adhere to 12pN FN-TGTs and are often balled up and rounded; thus though more TGTs are ruptured per cell spread area resulting in a brighter signal with 12pN TGTs than 54pN TGTs, the cumulative fluorescence over the entire cell area is lower for 12pN FN-TGTs. **Cumulative fluorescence intensity over the entire cell area is the analogous readout to our flow cytometry readout.** This has also been observed in many image based TGT studies (Supp Table 1). In contrast, cells spread equally well on 12 and 54pN TGTs with the echistatin ligand, which means the 12pN signal is larger both because of the more 12pN TGTs ruptured per cell AND the equivalent cell area. We have quantified this in Supplementary Fig 3.

Major concern: SI Fig 4B, SI Figure 5, SI Figure 6, and SI Figure 7 seem to be the result of single measurements and are not replicated and thus lacks statistical rigor. I may be mistaken but there is no error bars and no description of statistics in the captions or plots. SI figure 9 shows duplicates of experiments and in some cases triplicates but the caption fails to explain what colors and data points mean.

- We did defer to “representative plots” for some of the Supplemental figures because all of the data looked the same. We have now added in replicate data and included statistics.

Major concern: line 226 states “As expected, rupture of both 12 and 54pN TGTs was decreased upon treatment para-amino-Blebbistatin”. However SI Figure 11 is not consistent with this claim and the 54pN TGT did not show statistically significant decrease in signal following Bleb treatment.

- While the 54pN blebb treatment was not statistically significant ($p= 0.36$), there is a clear trend towards decreasing fluorescence with blebb treatment. The lack of statistical significance could be due to our stringent use of median fluorescence. Alternatively, U251 cells may not be as strongly affected by blebbistatin as other

cells, as it is known that microtubule dynamics also play a significant role in U251 traction force generation and migration⁵.

Minor issue: Line 236 states “We observed U251 cells bound 3-4 fold more than CHO-K1 cells, suggesting a greater number of integrin clutches available to exert traction force.” I think this conclusion is not fully correct. The greater bound probe on U251 cells may be the result of many other factors such as cell size, copy number of integrins displayed on the membrane, and also the competition between FN binding and Echi binding for each cell types.

- We have included a sentence that the signal could also be higher for U251 cells due to cell size and factors related to integrin concentration at the cell surface

Minor point: Figure 1b: It is surprising to see that CHO-K1 cells cultured on the 12pN TGT probes to spread when the ligand is Echi. This finding strongly suggests that the Echi ligand may irreversibly bind to integrins and also it may partially activate integrin receptors in the absence of tension. This response is cell line dependent as U251 cells showed lower cell spreading on 12pN Echi surfaces compared to that of the 54pN-Echi surfaces. The discussion needs to address this confusing point as it is an inherent quirk of using Echi as a ligand.

- Again, we believe that the broader integrin binding profile and higher affinity of the echistatin ligand supports adhesion on 12pN TGTs. We agree that this is a surprising quirk but would like to point out that relationship between spread area and surface composition is preserved between cell lines as seen in supplementary figure 9, in which U251 and CHO-K1 cells both show increases in spread area on 54 pN surfaces compared to 12 pN surfaces. We agree that the morphological differences between 12 and 54 pN surfaces is more robust with U251 cells and are excited to further investigate mechanisms behind this and echistatin supporting adhesion to 12pN TGTs.

Minor point: Fig. 1C: the accompanying brightfield images are absent from main figure and shown in SI. These are needed so that the readers can compare the location of the fluorescence signal with the location of the cell. It is very difficult to make sense of the data without knowing where cells are located and the overlap of the tension signal to the cell location. Another issue is that the fluorescence contrasts in SI figure 3 is different from that of figure 1c.

- Due to the already large size of Fig 1 and the availability of the side by side brightfield and fluorescence images in Supplementary Fig 3, we addressed this concern by overlaying the outlines of cells from brightfield images as white dotted lines. The contrast is not different, the panels in the main figure are zoomed in, perhaps making the images appear brighter.

Minor point: Descriptions such as “Streaks” and “puncta” around line 150 are vague and need better descriptions.

- The term “streak” has been used in the cited literature when describing the motile focal adhesions thus we believe this is a sufficient and accepted use of the term. Puncta were further elaborated on.

Minor point: The conclusion of SI Figure 6 is highly dependent on the size of area of the substrate so this information should be included.

- Thank you, we have made said improvements.

Minor typo: Pg 3. Line 118: change alpha5 to alphaV.

Reviewer #3 (Remarks to the Author):

The manuscript has been improved significantly with additional experiments and clarifications. I now understand that the main application of the technology will be in high throughput screen of mechanotypes using the most favorable ligand. The authors also present an early demonstration of 'screening based on mechanotypes' here using a mixture of cells and mixture of TGTs. I recommend publication.

1. I noticed that WDV is never defined, and sometimes used in lower case.

We have standardized and defined WDV.

2. The authors performed experiments to support the idea that cell fluorescence comes from internalized TGTs. The most direct demonstration would be to show that there is strong TGT fluorescence inside the cells based on microscopy data. It is curious why they do not present such data.

It is actually very difficult to distinguish the fluorescence inside the cell from the surface fluorescence using our widefield microscope. We have some fuzzy images from an attempt at confocal microscopy, but since intracellular fluorescence has been previously reported⁶ and the point of our assay was to provide an alternative to high resolution imaging, we did not expend a lot of effort on this.

References:

1. Seroussi, E. Estimating Copy-Number Proportions: The Comeback of Sanger Sequencing. *Genes* **12**, (2021).
2. Hu, Y. *et al.* DNA-Based Microparticle Tension Sensors (μ TS) for Measuring Cell Mechanics in Non-planar Geometries and for High-Throughput Quantification. *Angew. Chem. Int. Ed Engl.* **60**, 18044–18050 (2021).
3. Qi, L. *et al.* Talin2-mediated traction force drives matrix degradation and cell invasion. *J. Cell Sci.* **129**, 3661–3674 (2016).
4. Wang, Y., Zhao, Y., Sarkar, A. & Wang, X. Optical sensor revealed abnormal nuclease spatial activity on cancer cell membrane. *J. Biophotonics* **12**, e201800351 (2019).
5. Prael, L. S. *et al.* Microtubule-Based Control of Motor-Clutch System Mechanics in Glioma Cell Migration. *Cell Rep.* **25**, 2591–2604.e8 (2018).
6. Wang, X. *et al.* Integrin Molecular Tension within Motile Focal Adhesions. *Biophys. J.* **109**, 2259–2267 (2015).

REVIEWERS' COMMENTS

Reviewer #2 (Remarks to the Author):

It seems that the authors have addressed all my previous comments and I am satisfied with the updated manuscript.